

# Is groundwater sufficient to support sustainable irrigation agriculture in a reclaimed wetland region?

Zhonghe Pang[1], Lianghua Lv[1,2], Jie Li[1], Lijuan Yuan[1], Yanlong Kong[1], Lu Luo[1], Tianming Huang[1]

[1]Key Laboratory of Shale Gas and Geoengineering, Institute of Geology and Geophysics, Chinese Academy of Sciences,

Beijing, 100029, China

[2]University of Chinese Academy of Sciences, Beijing, 100049, China

*Correspondence to*: Zhonghe Pang (z.pang@mail.iggcas.ac.cn)

**Abstract.** Water resources management is the key to sustainable agriculture and wetlands ecosystems. Understanding whether agriculture with groundwater-dominated irrigation is sustainable is sometimes difficult due to complex hydrological conditions, e.g. in the case of a wetland region. To investigate this issue we have chosen a wetland and rice paddy fields co-existing area with a groundwater-dominated irrigation scheme in the Sanjiang Plain, NE China, which has been reclaimed from lake-swamp type of natural wetlands since the 1950s. Using a multi-tracer approach involving water chemistry and isotopes ($^2$H, $^{18}$O, $^3$H, $^{13}$C, $^{14}$C), integrated with data on groundwater regime, we demonstrate that it is possible to delineate the mechanism of hydraulic interactions between groundwater and river, ponds and rice paddy fields in the wetlands terrain. Regional variations in hydrogeology have been found to be the main factors controlling groundwater recharge and regime. Groundwater in the confined Quaternary aquifer with ages over 50 years and evidenced by depleted heavy isotopes is recharged by lateral flow from nearby mountains. The groundwater is in general not affected by surface activities, however, its yield is limited. Groundwater alone is not sufficient to support sustainable irrigation agriculture. On the contrary, the unconfined Quaternary aquifer is recharged by rainfall or riverbank infiltration, especially at localities near the rivers. It is more likely for the groundwater to be affected by agricultural activities, though its yield is rather abundant. This paper also indicates that utilization and planning of water resources for reclaimed agriculture can be improved through hydrological studies using environmental tracers.

**Keywords:** groundwater-surface water interaction; wetlands; sustainability of agriculture; isotopes; Sanjiang Plain



# 1 Introduction

Wetlands are ecologically and functionally significant elements of the water environment and have an important role in achieving sustainable river basin management by contributing to the abatement of the impact of pollution, as well as mitigating the effects of droughts and floods and increasing groundwater recharge. Nonetheless, wetlands continue to be drained and converted into agricultural land or industrial land, and as much as 50% of the world's wetlands have been lost (Assessment, 2005). Wetlands conservation is therefore of broad scientific interest. Although wetlands protection is officially a priority for the 159 nations that have ratified the Ramsar Convention (as of 2009), it is still very common to see member countries actively supporting wetlands reclamation as a part of their national agricultural policy (Scialabba, 1999). Nowadays, increasing populations continue to add pressure to the global food supply, and the total demand for food production is expected to rise 50% by 2030 (Hassan et al., 2005). Furthermore, the added stress of increasing cultivation for bio-energy plant production can be expected to accelerate the future trend of wetland reclamation.

Sustainable water supply and security in water quality are vital for sustainable agriculture in reclaimed wetlands. Groundwater is always the key factor that has allowed global agricultural production to expand in area and increase in output, and the consequential rise in pumping rates have resulted in groundwater level decline. For example, China has the world's largest population, and the policy of grain self-sufficiency brings high demand for agricultural irrigation water in northern China (Bradsher, 2011). Due to the overdraft of groundwater, there has been a significant drawdown of water levels in deep confined, semi-confined and/or unconfined aquifers, with an overall range of decline between 0.2 and 4 m/year in northern China (Currell et al., 2012).

To figure out whether groundwater is sufficient to support sustainable irrigation agriculture in a reclaimed wetland region requires an understanding of groundwater residence times, recharge mechanisms (Clark and Fritz, 1997; Scanlon et al., 2006) and groundwater's relationship to the wider water-cycle and surface water, including rivers, ponds and paddy fields. While wetlands hydrology is difficult to characterize (Hunt et al., 1998), environmental isotopes ($^{18}$O, $^{2}$H, $^{3}$H and $^{14}$C) of groundwater have been widely and extensively used to provide an effective means to identify water sources and trace their movement in wetlands (Hunt et al., 1998; Clay et al., 2004; Mills et al., 2011).

The Sanjiang Plain is the largest area of fresh water marsh and the largest food supply base in China. It is the largest

wetlands reclamation program in China and the world, for development of rice paddies. The total reclaimed area of the vast wetlands to paddy rice fields is approximately 4.5Mha during the past 50 years, accounting for 83% of the original wetlands area (Pan et al., 2011). In this study, with a focus on the groundwater-surface water interactions and implications on sustainable irrigation agriculture, we apply a multi-tracer approach for the first time to understanding groundwater recharge sources, residence times and factors controlling groundwater regime in the Sanjiang Plain wetlands terrain.

## 2 Study area

The Sanjiang Plain (43°49′-48°27′ N, 129°11′-135°05′ E) is located in the northeastern part of Heilongjiang Province, Northeast China (Fig. 1). It encompasses a total area of 6.2Mha. The Sanjiang Plain is bordered by the Xiaoxinganling Mountain to the southwest, and the Heilongjiang River and the Wusuli River to the north and east, respectively (Fig. 1). The elevation of the low plain ranges from 50 to 60m, while the elevation of the highest mountain is 1429m. The mean annual temperature increases from 1°C in the southern mountain region to 3°C in the northern plain (Liu and Ma, 2000). The mean annual precipitation in the plain is around 600mm, 70% of which falls between June and September.

In addition to the perennial Songhua River, there are many ephemeral rivers, e.g., the Nongjiang River and the Bielahonghe River, running through the Sanjiang Plain (Fig. 1). The thick Quaternary sediments deposited in this area are mainly alluvial, fluvial, or lacustrine sediments. In the western part of the Sanjiang Plain, the diluvial aquifer is formed of highly permeable cobble and gravel deposits, forming a uniform unconfined aquifer (District I) and the water tables are shallower than 10m. To the east, the aquifer is covered by a 16-20m thick clay and becomes confined or semi-confined (District II) (Fig. 2); in the lower reaches of the plain, the aquifer is formed of highly permeable cobble and gravel deposits, forming a uniform unconfined aquifer (District III).

The Sanjiang Plain contains a historically famous marsh, Bei Da Huang (Huang et al., 2010). In the 1940s, more than 5Mha of marshes and wet meadows existed (Liu and Ma, 2002). However, in order to meet the food demand spurred by the increasing population, reclamation plans called for agricultural land to be developed from wetlands in the Sanjiang Plain. Thereafter the cultivated land area has increased from about 0.79Mha in 1949 to 5.24Mha in 2000, while the wetlands area has decreased from 5.35Mha in 1949 to 0.84Mha in 2000 (Liu and Ma, 2000; Li et al., 2002). Paddy cultivation dominates





the agricultural sector, leading to large amount of groundwater exploitation (6.65mega $L \cdot ha^{-1} \cdot yr^{-1}$ (corresponding to 665mm) and fertilizer application (170kg $N \cdot ha^{-1} \cdot yr^{-1}$). Urea ($CO(NH_2)_2$), diammonium phosphate (($NH_4)_2HPO_4$) and ammonium bicarbonate ($NH_4HCO_3$) are the most widely used nitrogen fertilizers, and are usually applied in April.

## 3 Sampling and analyses

Groundwater samples, as well as surface water in paddy fields, drainage channels, and rivers, were taken for isotopic ($^2H$, $^{18}O$, $^3H$, $^{13}C$ and $^{14}C$) and chemical analyses from three typical farms (HH, QF and QS Farm, respectively) at the northeast part of the Sanjiang Plain in July 2009. Following preliminary interpretation of these data, a further sampling campaign for isotopes was conducted in August 2011 along a transect extending 250km in east-west direction across the plain. Two typical hydrogeology conditions can be found along the transect with the unconfined aquifer in the west and confined aquifer in the east (Fig. 2). Precipitation samples from the Sanjiang station were collected monthly between January 2005 to November 2006 (data from China precipitation isotope network) and August 2010 to September 2011 (conducted by this study). Locations of all samples are shown in Fig. 1.

All water samples for chemical analysis were filtered with a 0.45μm membrane. An aliquot was acidified with 1% $HNO_3$ for cation analysis. Water chemistry was measured at the Beijing Research Institute of Uranium Geology. The cation measurement was based on National Analysis Standard DZ/T0064.28-93 while anions based on DZ/T0064.51-93. Analytical precision was 3% of concentration based on reproducibility of samples and standards. Stable isotopes were analyzed using PICARRO.L1102-i Laser Absorption Water Isotope Spectrometer in the Water Isotope Lab of Institute of Geology and Geophysics, Chinese Academy of Sciences. Results are reported as $\delta^2H$ and $\delta^{18}O$ ($\delta = (R_{sample}/R_{standard}-1) \times 1000$) with the standard of Vienna Standard Mean Ocean Water (VSMOW). The analytical precision is 0.5‰ for $\delta^2H$ and 0.1‰ for $\delta^{18}O$.

To examine the ages of groundwater, samples for $^3H$ and radiocarbon were collected. Tritium was determined on electrolytically enriched water samples by low-level proportional counting and the results are reported as tritium unit (TU) with a typical error of 1TU. The measurement was performed at the Open Laboratory of Environmental Geology and the Central Laboratory of Hydrogeology, the Ministry of Land Resources, China. $^{14}C$ of dissolved inorganic carbon (DIC) was determined radiometrically by liquid scintillation counting after conversion to benzene in Beta Analytic Inc in the USA. The

specific $^{14}$C activity was reported as percent modern carbon (pmC).

Geochemical results (site information, field data, stable isotopes of waters and $^3$H) are shown in Table 1.

In order to obtain the local meteoric water line (LMWL) for Sanjiang region, we have used data of stable isotopes in precipitation sampled at the Honghe Station during January to December 2005 and during August 2010 to July 2011,

respectively. The LMWL was constructed by plotting δD (deuterium) versus δ$^{18}$O and the equation was obtained through linear regression. In order to obtain the local evaporation line, we used the stable isotopes in surface water (rivers and channels) of the region to perform a linear regression analysis. Based on the evaporation line, we further obtained the mean isotope value of the local precipitation, which is equal to that of the intersection point on the LMWL with the evaporation line. The local evaporation line will be used as a reference line in distinguishing if groundwater has been recharged by

surface water.

In order to interpret Tritium ($^3$H) data to obtain an estimate of groundwater age, we used the GNIP (IAEA/WMO Global Network of Isotopes in Precipitation) station at Qiqihar (latitude 47°23'0"). The limited record of Tritium monitoring was further extended making use of that of GNIP station at Ottawa, Canada (latitude 45°23'0"), based on the principle of latitude effect.

In order to determine the initial $^{14}$C content in the atmosphere, we plotted $^3$H content against $^{14}$C activity. When $^3$H value is below the $^3$H detection limit, or in other words, the water is Tritium-free, the corresponding $^{14}$C activity can be considered to be the initial $^{14}$C value, which was then used to deduce the $^{14}$C age on the basis of the $^{14}$C decay law.

There are two kinds of hydrogeological units in the Sanjiang Plain, as detailed in the "Study area" section of this paper. In the western part (District I) and the northern part (District III) of the study area, the aquifers are unconfined that are

composed of highly permeable cobble and gravel deposits. In contrast, in the eastern part (District II), the aquifer is confined or semi-confined and is covered by a 16-20m thick clay layer. We used data on groundwater regime to demonstrate quantitatively the impact of such differences on groundwater recharge. Our current efforts are aimed to provide evidences from hydrogeochemical and isotopic tracers.

**4 Results**



## 4.1 Ca$^{2+}$ and NO$_3^-$ content

Figure 3 illustrates that the concentrations of groundwater NO$_3^-$ vary greatly under different hydrogeological conditions: in the phreatic aquifer (Districts I and III), the concentrations of NO$_3^-$ range from <0.05 to 458mg/L; while in the confined aquifer (District II), most of them are less than 10mg/L. The groundwater Ca$^{2+}$ behaves similarly to groundwater NO$_3^-$ (Fig. 4). An elevated calcium concentration (>80mg/L) can be found in the shallow groundwater of the phreatic aquifer (Districts I and III), while in the confined aquifer (District II), most of them are lower (<80 mg/L).

## 4.2 Isotopes in the precipitation

The isotopic values in the Sanjiang station and the corresponding amount of precipitation in each month were listed in Table 2. The δ$^{18}$O values vary from -28.2‰ to -4.7‰ and δ$^2$H from -207.3‰ to -38.26‰. Precipitation during the winter season is more depleted than that of the summer season. By November of every year, the air temperatures begin to drop below zero and the isotopic compositions of precipitation in the form of snow are always lower than -15‰ and -100‰ for δ$^{18}$O and δ$^2$H, respectively. This is in accordance with data records from Qiqihar Station of the global network for isotopes in precipitation (GNIP). The average annual weighted mean of δ$^{18}$O and δ$^2$H in the Sanjiang Station for the two complete years (from January 2005 to December 2005 and from August 2010 to July 2011) are –12.3‰ and -90.7‰, respectively.

A significant linear correlation (R$^2$ = 0.99) exists between the two parameters δ$^{18}$O and δ$^2$H (Fig. 5). Both slope (7.51) and intercept (-0.92) of the local meteoric water line (LMWL) show a typical isotopic pattern of precipitation in a cold region (Craig, 1961; Rozanski et al., 1993).

## 4.3 Isotopes in surface water

Fifteen samples of surface water were collected in the study area in order to distinguish the isotopic signature of rivers and irrigation water from paddy fields. The river water samples S1 and S4 are plotted on the LMWL. However, samples S2 and S3 are located along a line with a slope of 5.7, indicating the evaporative enrichment effect (Fig. 6). The channel water samples (C1 and C2) are enriched in heavy isotopes, and also show the effect of evaporation. The isotopic composition of irrigation water from rice paddy fields entering drainage channels is also comparable to that of rainfall, showing a wide

range from -9.0‰ to -12.6‰ for $\delta^{18}O$, with a mean value of -10.4‰. Most of the irrigation water in paddy fields lies above the LMWL, which may be ascribed to the effect of condensation.

The tritium contents of the two surface water samples, i.e. Bielahonghe River (S4) and wetlands in the HH Farm are $25.1\pm2.0$ and $22.4\pm2.0$ TU, respectively (Table 1).

### 4.4 Isotopes in groundwater

Stable isotopes of groundwater samples collected from the unconfined area (districts I and III) and the confined area (district II) along the transect A-A' display significant variations. In the confined area, all the samples are located near the LMWL except for DX16 (Fig. 7). Stable isotopic composition ranges from -8.7‰ to -12.6‰ with a mean value of -11.6‰ for $\delta^{18}O$,

and from -73.5‰ to -92.0‰ with a mean value of -86.9% for $\delta^{2}H$. Whilst in the unconfined area, stable isotopic compositions show a little enrichment with a mean value of -11.1‰ and -84.4‰ for $\delta^{18}O$ and $\delta^{2}H$, respectively (Fig. 7).

Groundwater in the three farms (HH, QF, and QS Farm) have specific isotopic characteristics (Fig. 8). Oxygen isotope compositions of groundwater samples at HH and QF Farms range from -13.5 to -12.2‰ and -13.8 to -12.3‰, with averages of -12.9‰ and -13.0‰, respectively. The values of $\delta^{18}O$ and $\delta^{2}H$ from four groundwater samples collected at QS farm are

15 higher than that of others, whose values are closer to that of the surface water.

Tritium contents of groundwater were measured only for the samples from the three farms, representing confined and unconfined areas (Table 1). The tritium contents of groundwater from the HH and QF Farms in the confined area (district II) range from <1.0TU to 2.2TU with groundwater depth ranging from 18 to 120m. However, tritium contents of groundwater from the QS Farm in the unconfined area (district III) range from <1.0 to 71.3TU with groundwater depth ranging from 20 to

20 72m.

Groundwater $^{14}C$ was measured for samples selected from the three districts (Table 3). Samples FJ02, FJ14, HC03, HG01, HG10 and QS1 represent the unconfined aquifer, and DX02, DX12, HH1, HH2, QF1 the confined aquifer. The highest value is found in the shallowest well (sample QS1 with well depth of 20m) and lowest value is found in the deepest well (sample FJ14 with well depth of 152m).



## 5 Discussion

### 5.1 Recharge to aquifers

Stable isotope data can be used to evaluate possible sources of water to aquifers. Groundwater collected from the unconfined area (District I) and the confined area (District II) along the transect A-A' displays significant differences in isotopic composition (Fig. 7). Most groundwater in the confined area is located on the LMWL, while some groundwater in the unconfined area is located on the local evaporation line. Isotopic compositions of groundwater in the unconfined area are more enriched than that of groundwater in the confined area. This suggests that groundwater in the unconfined area is more easily recharged by evaporated surface water with more enriched isotopic compositions.

Nitrate concentration in groundwater is used as a primary indicator of agricultural impact (Edmunds, 2009). In District I, phreatic groundwater nitrate concentrations can reach 458mg/L, while in the confined aquifer, nitrate in most of the samples is less than 10mg/L (Fig. 3). This suggests that groundwater in the phreatic aquifer is affected more remarkably by surface water via vertical infiltration than that in the confined aquifer, which is further confirmed by the distribution of $Ca^{2+}$. It is interesting that groundwater $NO_3^-$ concentrations in District III are also less <10mg/L, which is quite different from that of District I. One of the reasons is that fertilizers were used less in District III than in District I.

The groundwater $Ca^{2+}$ behaves similarly to groundwater $NO_3^-$. An elevated calcium concentration (>80mg/L) can be found in the shallow groundwater of the phreatic aquifer (Districts I and III), while in the confined aquifer (District II), most of them are lower (<80mg/L) (Fig. 4). $Ca^{2+}$ contents of the soil in Sanjiang Plain are high with the effective calcium content between 2500-4500mg/L (Li et al, 2010). The unconfined aquifer is influenced by vertical infiltration, and the interaction with the calcium-rich soil leads to the high calcium concentrations in the shallow groundwater. However, in the confined aquifer, lateral groundwater flow dominates the groundwater recharge. Lack of carbonate are certified by the low $^{13}C$ values of groundwater (Table 3), so no high calcium concentrations of groundwater are found in District II.

Isotopic compositions of groundwater from three farms (HH, QF, QS) are used to identify the groundwater-surface interaction and the $\delta^2H$-$\delta^{18}O$ plots of groundwater are shown in Fig. 8. The aquifers of HH and QF farms are confined (District II) and that of QS is partly unconfined (District III). The groundwater samples at HH and QF farms are more depleted in heavy isotopes than the surface water, further indicating that lateral groundwater flow dominates the groundwater

recharge, and that interaction between groundwater and surface water does not occur. This is in accordance with groundwater levels at HH farm which show similar changes with overall water table decline but with intra-annual fluctuations observed with the groundwater exploitation.

A distinctive isotopic feature is found at the QS farm. Groundwater is enriched in $^{18}O$ as compared to that of HH and

QF farms. This enrichment suggests that the aquifer at QS farm is influenced by vertical infiltration, and there exists a relatively strong connection between groundwater and the surface water. QS Farm is located close to the Bielahonghe River (S3 with $\delta^{18}O$ of -9.6‰ and $\delta^2H$ of -77.9‰), and groundwater levels decline only moderately with negligible intra-annual fluctuation (Fig. 8).

**5.2 Residence time of groundwater**

Tritium is one of the most important transient and ideal tracers used in hydrological research as it is incorporated into, and carries information along, the water molecule itself (Michel, 2005). Tritium levels recorded near the study area, at the Qiqihar GNIP station (latitude 47°23'0"), show a similar range to levels in Ottawa, Canada (latitude 45°23'0") (Brown, 1961), during an overlapping period (Fig. 9). Both of the tritium records were taken from the IAEA network (http://isohis.iaea.org).

Based on the principle of latitude effect, the precipitation tritium record of Ottawa can be regarded as the input function of tritium in precipitation in the Sanjiang Plain.

The current atmospheric level of tritium is around 10TU. Groundwater recharged before the nuclear testing should have tritium below 6.5TU. Samples with tritium values higher than this value are considered to be recharged after the nuclear testing, or approximately 1960. Tritium in groundwater from HH and QF farms show low levels, with narrow ranges of

<1.0-1.9TU and <1.0-2.2TU, respectively, indicating that groundwater at HH and QF farms is older than 50 years. Groundwater at QS farm shows a wide range of tritium levels (<1.0-71.3TU), corresponding to the different sampling locations. Samples with high levels of tritium (6.5-71.3TU) are from shallow groundwater collected near the river, and those collected away from the river are from deeper groundwater showing low levels of tritium (<1.0TU). For example, the $^3H$ concentration of 29.9TU in sample QS2, clearly indicating that there is a young (post-bomb-peak) component derived most

probably from river water. This location dependent tritium concentrations suggest that groundwater near the river has



relatively short residence times. The distribution of tritium in the three farms suggests that the groundwater in the confined area is pre-modern while that in the unconfined area is recharged by modern water near the river where better hydraulic connections are developed.

Radiocarbon analyses for 11 groundwater samples are shown in Table 3 as percent modern carbon (pmC) with a range from 30.7 to 98.4pmC. The $^{13}C$ values are in the range of -13.5‰ and -20.1% (Fig. 10), indicating that they are less affected by carbonate dissolution and that the silicate weathering is the dominant process (Edmunds et al., 2006).

However, accurate $^{14}C$ age estimation depends on the knowledge of initial $^{14}C$ content, geochemical system and data availability. Plotting the $^{3}H$ value against $^{14}C$ activity can give a good indication of initial activity (Verhagen et al., 1974). When $^{3}H$ value is below the $^{3}H$ detection limit, the corresponding $^{14}C$ content can be considered as the initial value. This method could reduce uncertainties of initial $^{14}C$ estimates during recharge processes affected by temperature, pH, $CO_2$ partial pressure, soil context, and the effects of plant photosynthesis cycles on $^{13}C$ content. From such an analysis (Fig. 10), an upper limit of the initial $^{14}C$ activity of about 80pmC is obtained. Using this value, the ages of groundwater were determined by the $^{14}C$ dating method, and the results are shown in Table 3. Generally, the $^{14}C$ ages of groundwater range from modern to 7910 years. Groundwater at similar depths is older in the confined aquifer than in the unconfined aquifer, which is due to recharge from precipitation to the unconfined aquifer.

## 5.3 Implications on agricultural sustainability

Though groundwater in the unconfined area is recharged by modern water with strong interaction with surface water, it is more easily contaminated by agricultural activities. Compared with the elevated nitrate concentrations in some shallow groundwater samples, the significantly low levels of nitrate in river water (S4) indicate that the natural water quality enhancement has been performed by the natural wetlands (Zedler, 2003; Mander et al., 2005), which is defined as "wetlands function" (Maltby, 2009). From a sustainability perspective, reclamation of wetlands for agriculture near the river should therefore be strongly discouraged in order to increase the wetlands function, otherwise the groundwater will continue to be deteriorated.

Groundwater recharge in the confined area is dominated by lateral flow and renewal of groundwater is in the time scale

of thousands of years. So Integrated use of groundwater and surface water is the ideal solution for the agriculture in areas within district II, where local recharge to groundwater is rather limited and water table decline is expected to grow fast with the current irrigation practices. Groundwater regime needs to be monitored regularly in order to ensure a better management scheme for groundwater use.

## 6 Conclusions

Stable isotopes, tritium, radiocarbon and water chemistry data have enabled clarification on the interaction between groundwater and surface water in the wetlands ecosystem under agricultural transformation, using Sanjiang Plain, NE China as an example.

Results show that the interaction patterns between groundwater and surface water vary tremendously over the region as hydrogeological conditions change. High tritium, nitrate and evaporative isotopic signature in the unconfined aquifer suggest that surface water can easily infiltrate into shallow groundwater in districts I and III. On the contrary, in the confined aquifer of district II, where groundwater resources are derived from lateral flow, renewal of groundwater is in the time scale of thousands of years. Local vertical recharge to groundwater is limited and fast decline of the water table is anticipated with

the current irrigation practices.

       Our results have significant implications on wetlands conservation and agricultural sustainability in a wetlands terrain. More strict measures are needed to protect groundwater quality in localities of districts I and III, where high nitrate concentrations are found. For district II, integrated use of surface water together with groundwater is the optimal solution to the fast decline of water table and to ensure water supply for agricultural irrigation, while groundwater alone is not sufficient

to support sustainable irrigation agriculture.

### Acknowledgments

The research is supported by the International Atomic Energy Agency (RC15397), and the National Science Foundation of China (Grants 40872162 and 41202183). Thanks are due to Prof. Baixing Yan from Northeast Institute of Geography and

Agroecology, Chinese Academy of Sciences, for assistance during the field work. The earlier version of the manuscript was



reviewed and corrected by Prof. M. Edmunds, whose comments have considerably improved the quality of the paper.

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



**Figure 1. Geographical distribution of mountains, rivers and sampling sites in the Sanjiang Plain.**



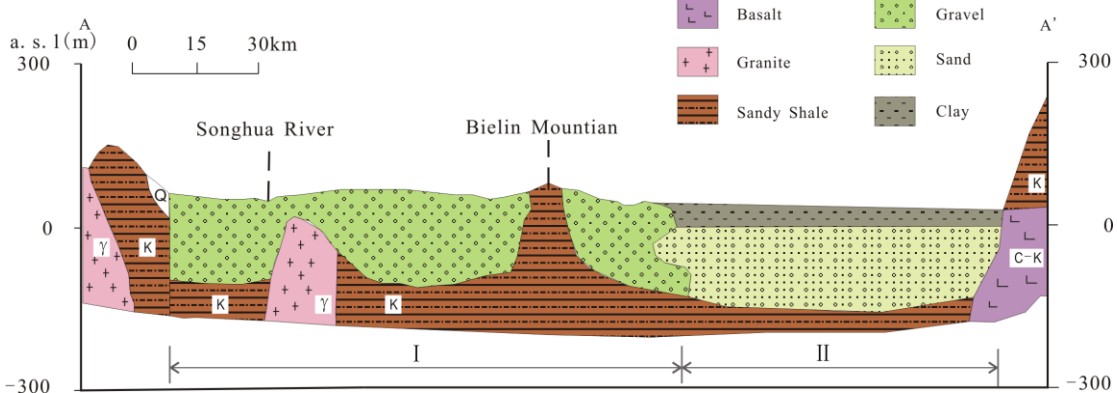

**Figure 2. Schematic hydrogeological map along the transect A-A' in Fig. 1**

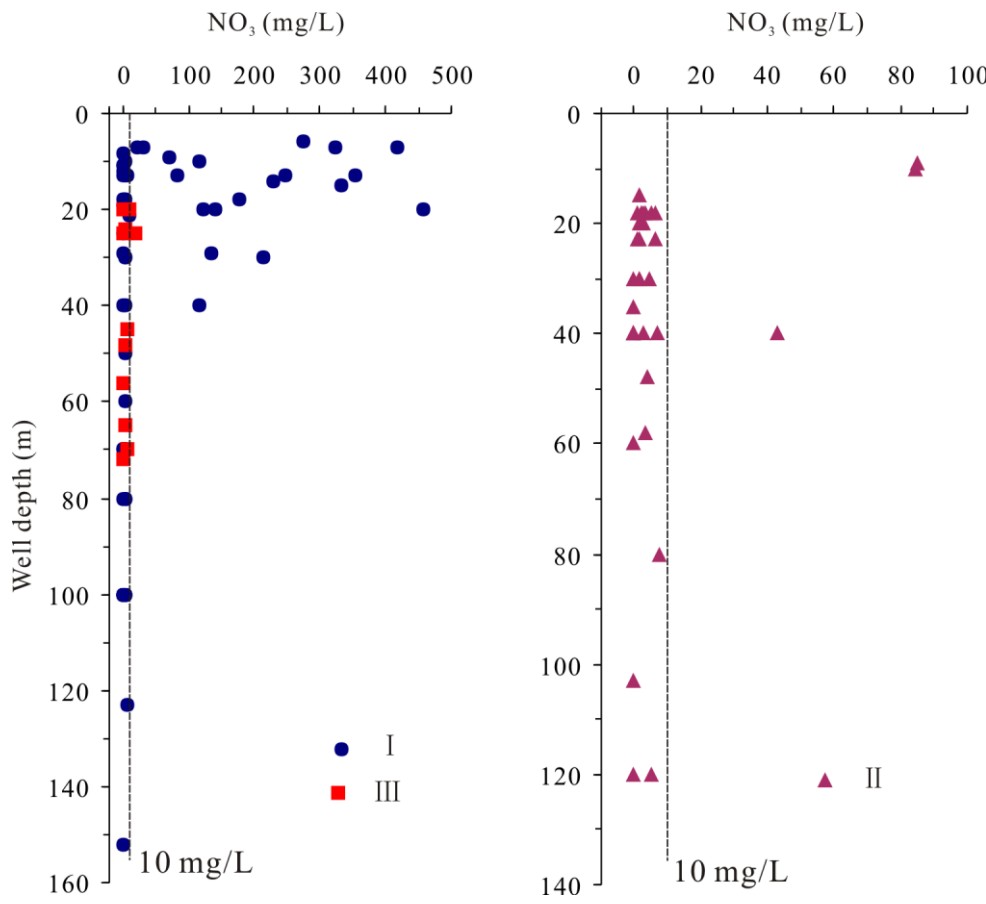

5    **Figure 3. Nitrate versus depth of groundwater from the three districts**




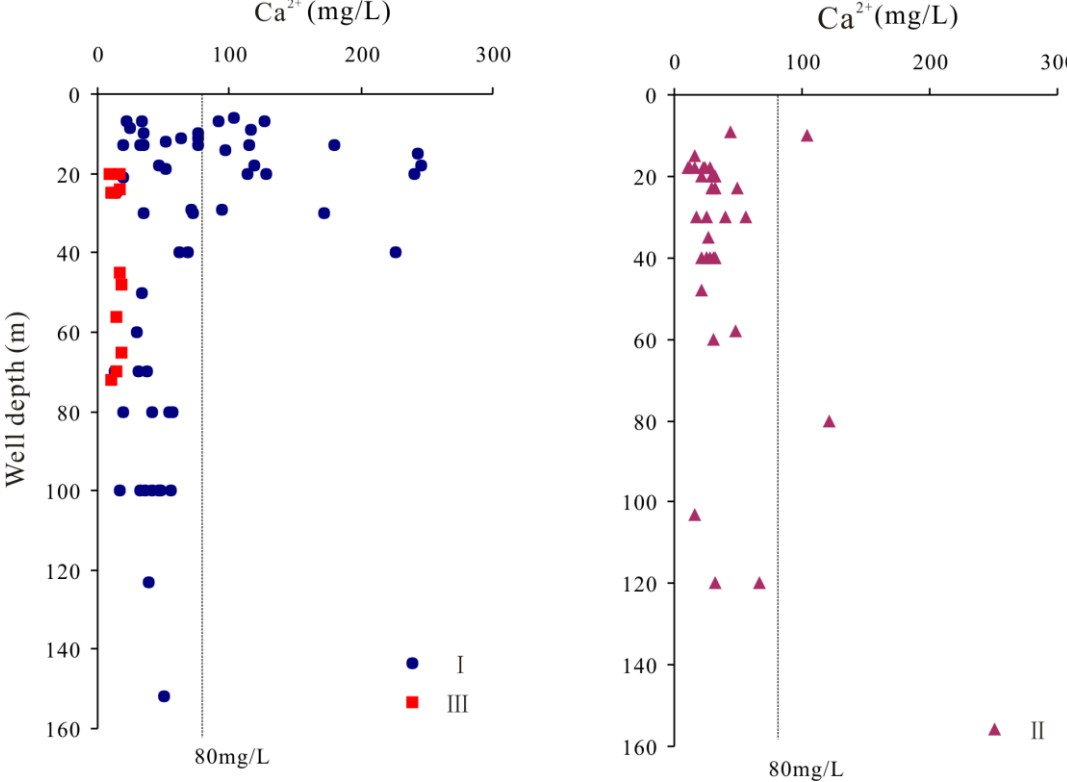

**Figure 4. Ca versus depth of groundwater from the three districts**

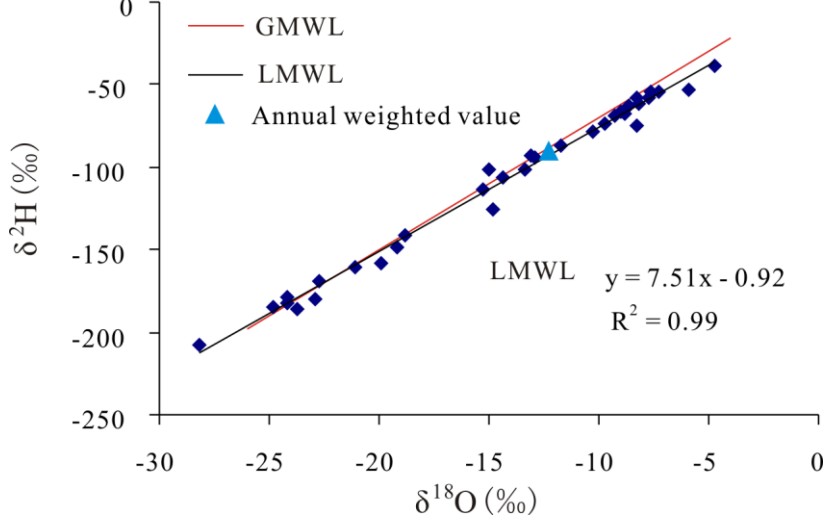

5      **Figure 5. Stable isotope plots of precipitation. Regression equation of LMWL is δ²H=7.51δ¹⁸O-0.92 (R²=0.99).**





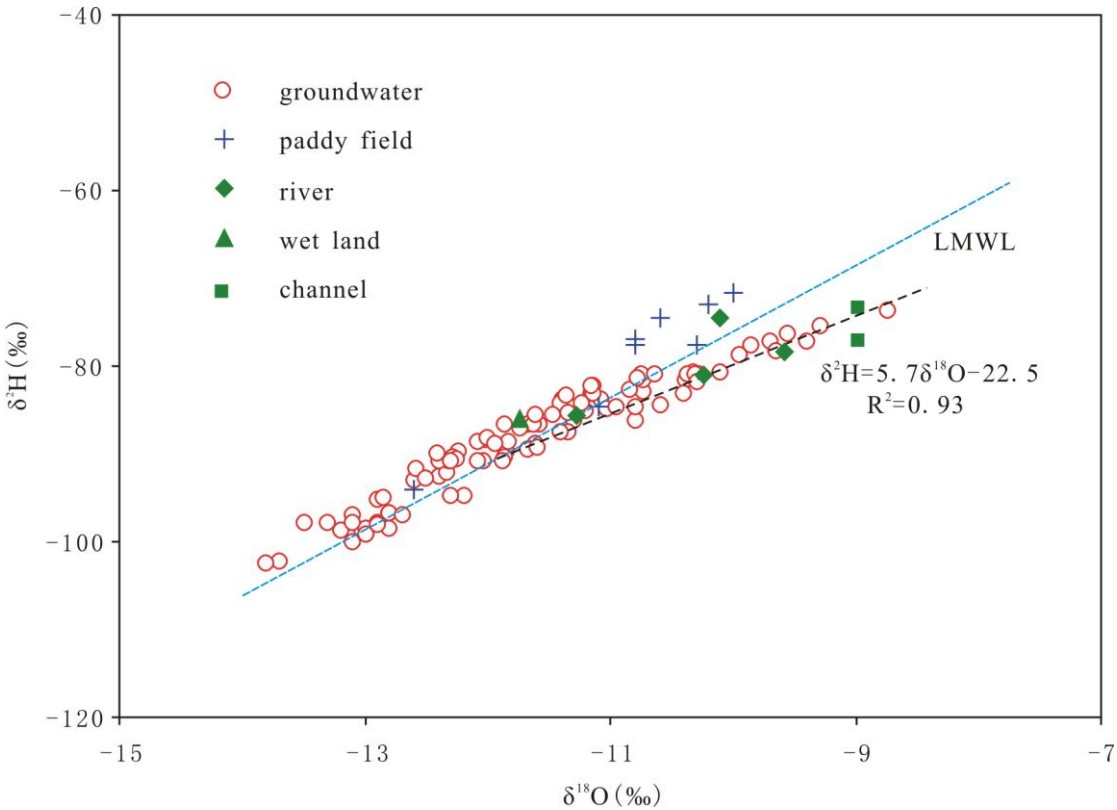

**Figure 6. The relationship between δ¹⁸O and δ²H of surface water and groundwater in the Sanjiang Plain**

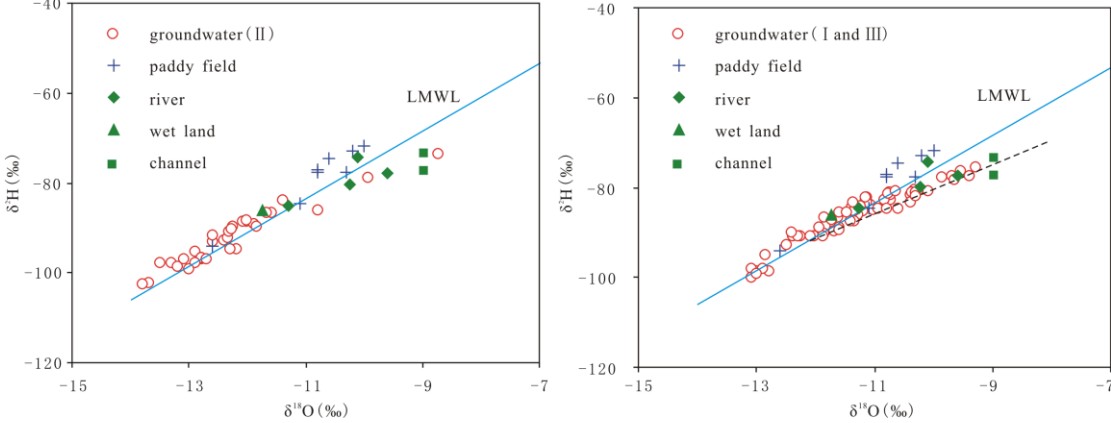

5      **Figure 7. The relationship between δ¹⁸O and δ²H of groundwater in the unconfined (District I and III) and the confined aquifer**

         **(District II) in the Sanjiang Plain**





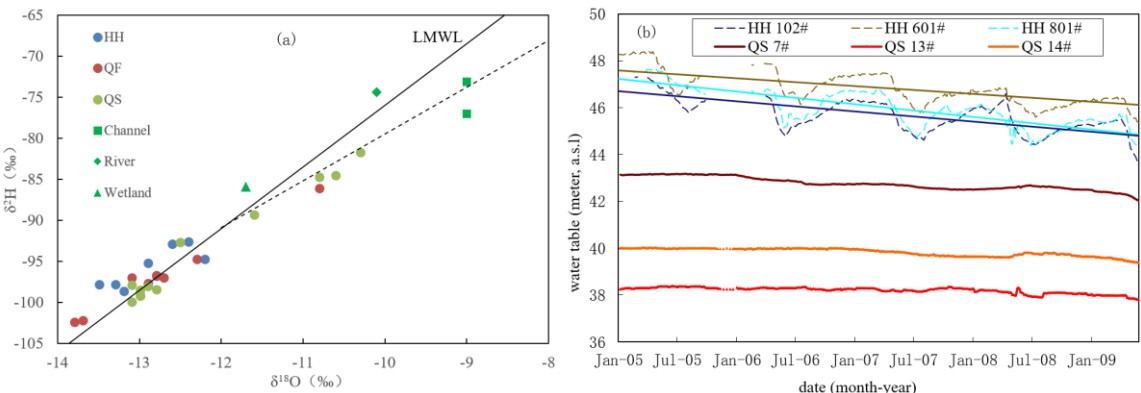

**Figure 8. (a) Isotopes of groundwater in Honghe (hereafter HH), Qianfeng (hereafter QF) and Qianshao farms (hereafter QS); (b)**

**Groundwater regime of the HH and QS farms.**

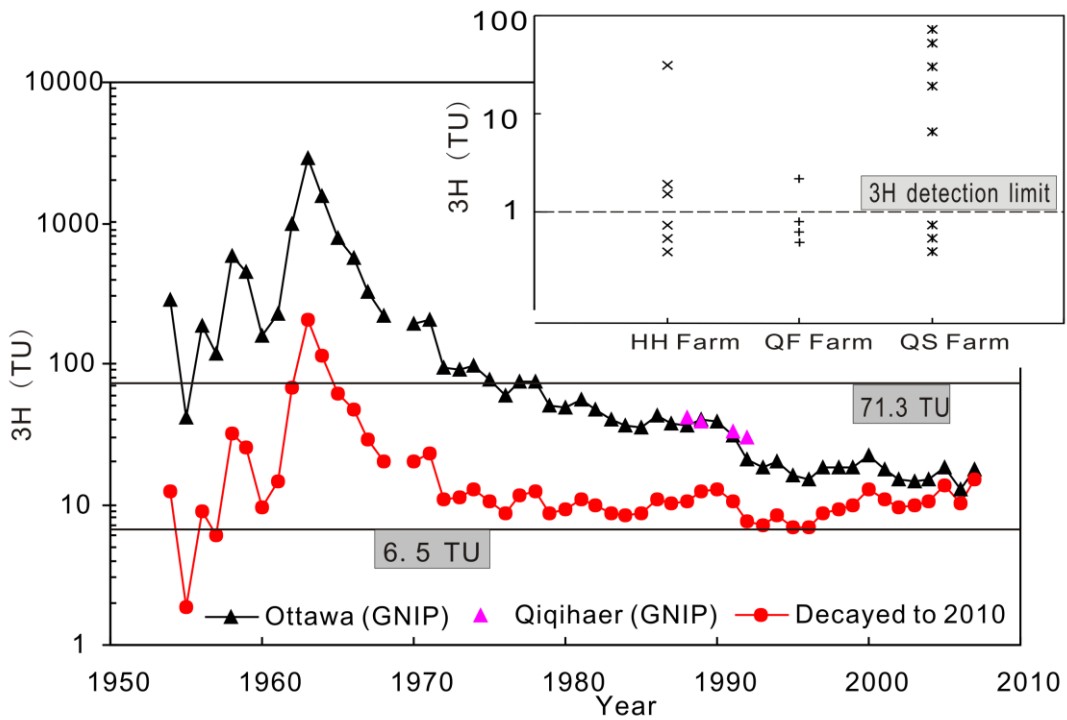

**Figure 9. Tritium in groundwater and precipitation from 1954 to 2007 and that decayed to 2010.**




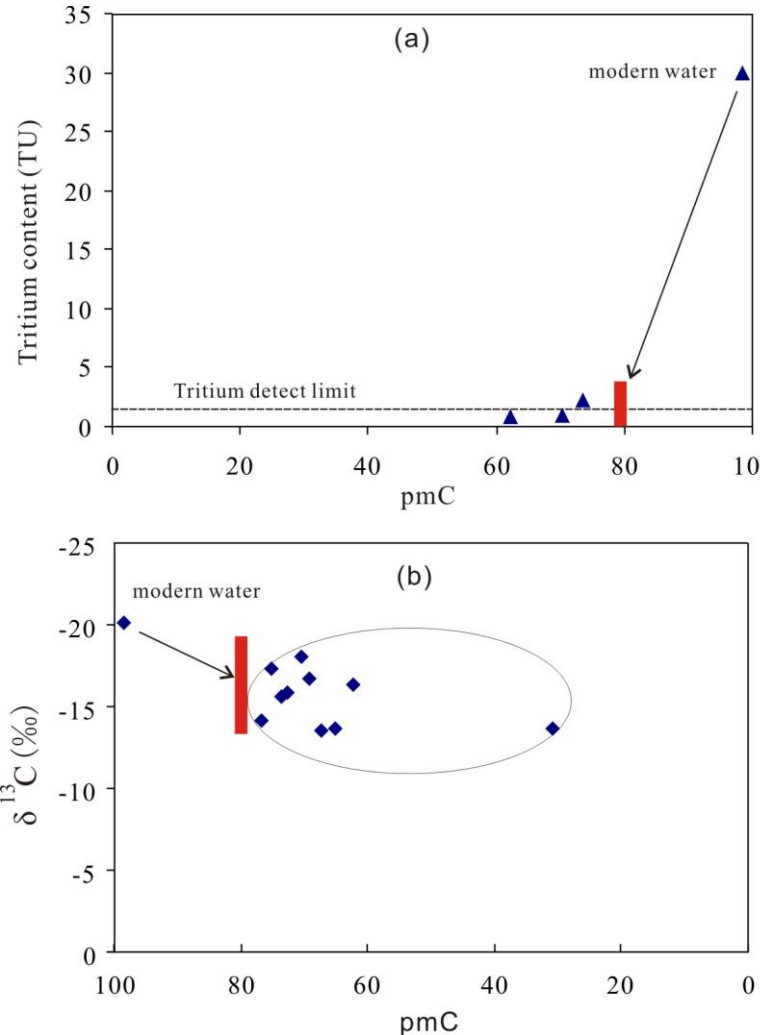

**Figure 10. (a) ³H content versus ¹⁴C content in groundwater. Once below the ³H detection limit, the ¹⁴C**

**content can be considered as the initial value; (b) The ¹³C content versus ¹⁴C content in groundwater.**



1 **Table 1 Chemical and isotopic compositions of groundwater and surface water samples in the Sanjiang Plain**

| Code | latitude | longitude | Depth | T | EC | pH | NO$_3^-$ | Ca$^{2+}$ | $\delta^{18}$O | $\delta^2$H | $^3$H |
|---|---|---|---|---|---|---|---|---|---|---|---|
| | | | (m) | °C | (us/cm) | | (mg/L) | | (‰) | (‰) | (TU) |
| Groundwater-District I-unconfined | | | | | | | | | | | |
| HG01 | 47.344 | 130.438 | 123 | 17.9 | 488 | 7.54 | 5.8 | 39.1 | -11.7 | -86.7 | nd |
| HG02 | 47.345 | 130.439 | 18 | 4.8 | 2800 | 6.73 | 0.8 | 246 | -11.1 | -83.8 | nd |
| HG03 | 47.345 | 130.438 | 9 | 11.9 | 741 | 6.87 | 71.8 | 117 | -11.2 | -83 | nd |
| HG04 | 47.309 | 130.552 | 70 | 9 | 131.9 | 6.92 | 2.6 | 13.5 | -12.4 | -90.7 | nd |
| HG05 | 47.306 | 130.551 | 21 | 9.7 | 223 | 6.78 | 8.7 | 19.2 | -12.3 | -90.7 | nd |
| HG06 | 47.255 | 130.579 | 25 | 10.5 | 263 | 6.35 | 0.9 | 14.2 | -11.6 | -86.6 | nd |
| HG08 | 47.228 | 130.769 | 70 | 6.9 | 422 | 6.75 | <0.05 | 30.8 | -9.7 | -77.2 | nd |
| HG09 | 47.226 | 130.768 | 13 | 14.1 | 310 | 7.09 | 0.6 | 34.7 | -10.7 | -82.8 | nd |
| HG10 | 47.211 | 130.815 | 100 | 8.4 | 187.4 | 6.83 | 2.6 | 16.6 | -11 | -84.9 | nd |
| HG11 | 47.204 | 130.816 | 7 | 9.3 | 1464 | 6.64 | 417 | 127 | -11.3 | -84.3 | nd |
| HG13 | 47.244 | 130.737 | 13 | 11.4 | 455 | 6.25 | 4.1 | 32.7 | -9.9 | -77.7 | nd |
| HG14 | 47.347 | 130.245 | 18 | 7 | 960 | 6.45 | 178 | 119 | -11.4 | -84.2 | nd |
| HG15 | 47.346 | 130.244 | 29 | 9 | 852 | 6.96 | 133 | 71.5 | -11.1 | -82.1 | nd |
| HC02 | 47.165 | 130.882 | 7 | 11.6 | 374 | 6.41 | 30.3 | 21.6 | -9.3 | -75.5 | nd |
| HC03 | 47.167 | 130.923 | 100 | 10.3 | 362 | 6.94 | <0.05 | 41.6 | -10.4 | -81.5 | nd |
| HC04 | 47.171 | 131.1 | 7 | 9.2 | 414 | 6.27 | 22.2 | 33.5 | -10.7 | -81.6 | nd |
| HC05 | 47.121 | 131.112 | 100 | 8.4 | 270 | 7.19 | 1.1 | 31.7 | -11.8 | -88.6 | nd |
| HC06 | 47.12 | 131.291 | 100 | 9 | 373 | 6.99 | 0.7 | 48.3 | -10.1 | -80.6 | nd |
| HC07 | 47.119 | 131.289 | 12 | 6.7 | 433 | 6.78 | 0.1 | 51.7 | -10.8 | -82.6 | nd |
| HC08 | 47.139 | 131.351 | 70 | 7.4 | 401 | 6.72 | <0.05 | 38 | -9.7 | -78.2 | nd |
| HC09 | 47.139 | 131.352 | 10 | 6.9 | 572 | 6.64 | 116 | 75.8 | -10.7 | -81 | nd |
| HC10 | 47.101 | 130.852 | 6 | 10.6 | 1449 | 5.86 | 276 | 104 | -10.8 | -81.3 | nd |
| HC11 | 47.089 | 130.91 | 80 | 10.1 | 270 | 6.7 | 1.2 | 20 | -10.3 | -80.6 | nd |
| HC12 | 47.112 | 130.932 | 8.5 | 11.2 | 280 | 6.57 | 0.6 | 24.7 | -9.6 | -76.2 | nd |
| HC13 | 47.121 | 130.97 | 100 | 10.8 | 367 | 6.94 | 2.1 | 35.8 | -10.4 | -83.2 | nd |
| HC14 | 47.12 | 130.969 | 7 | 9.9 | 1078 | 6.55 | 324 | 91.6 | -11.1 | -83.1 | nd |
| HC15 | 47.113 | 131.011 | 13 | 9.3 | 240 | 6.69 | 5.9 | 19.9 | -11.2 | -85.1 | nd |
| FJ01 | 46.904 | 131.506 | 30 | 9.8 | 289 | 7.21 | 2.1 | 35.1 | -12 | -90.7 | nd |
| FJ02 | 46.954 | 131.394 | 60 | 7.7 | 246 | 6.92 | 2.3 | 29.2 | -11.9 | -90.2 | nd |
| FJ03 | 46.98 | 131.414 | 50 | 7.9 | 290 | 7.14 | 2.8 | 34 | -11.9 | -90.8 | nd |
| FJ04 | 47.004 | 131.45 | 10 | 9.3 | 294 | 6.93 | 3.8 | 34.6 | -11.6 | -88.8 | nd |
| FJ05 | 47.031 | 131.482 | 18 | 7.6 | 356 | 6.86 | 3.3 | 46.6 | -11.3 | -86.1 | nd |
| FJ06 | 47.037 | 131.443 | 11 | 7.4 | 546 | 6.92 | 1.7 | 64 | -9.4 | -77.2 | nd |
| FJ07 | 47.08 | 131.457 | 30 | 6.9 | 577 | 6.93 | 2.1 | 72.5 | -10.4 | -80.9 | nd |
| FJ08 | 47.09 | 131.503 | 40 | 7.9 | 533 | 6.89 | 3.1 | 62.6 | -10.3 | -80.8 | nd |
| FJ09 | 47.082 | 131.412 | 40 | 9.2 | 491 | 7.09 | 1.3 | 68.5 | -10.3 | -80.9 | nd |
| FJ10 | 47.081 | 131.413 | 11 | 5.5 | 653 | 7.1 | 1.2 | 76 | -11.3 | -87.5 | nd |
| FJ11 | 46.98 | 131.665 | 40 | 8 | 1452 | 7.27 | 117 | 226 | -11.9 | -88.7 | nd |





| | | | | | | | | | | | |
|---|---|---|---|---|---|---|---|---|---|---|---|
| FJ12 | 46.981 | 131.667 | 15 | 10.5 | 1300 | 7.16 | 333 | 243 | -11.7 | -87.1 | nd |
| FJ13 | 47.039 | 131.647 | 30 | 10.2 | 1112 | 6.93 | 214 | 172 | -13 | -98.6 | nd |
| FJ14 | 47.04 | 131.648 | 152 | 10.5 | 408 | 7.79 | 0.1 | 50.4 | -12.9 | -95 | nd |
| FJ15 | 47.05 | 131.752 | 100 | 11.1 | 319 | 7.46 | 1 | 46.6 | -12.3 | -90.7 | nd |
| FJ16 | 47.051 | 131.752 | 13 | 10 | 561 | 6.66 | 84 | 75.9 | -11.6 | -85.5 | nd |
| FJ17 | 47.034 | 131.76 | 13 | 9.5 | 1131 | 6.87 | 353 | 180 | -10.6 | -80.8 | nd |
| FJ18 | 47.034 | 131.762 | 20 | 11.3 | 788 | 6.73 | 141 | 114 | -11.2 | -84.1 | nd |
| FJ19 | 47.025 | 131.966 | 20 | 9.6 | 1866 | 6.67 | 458 | 240 | -11.3 | -85.2 | nd |
| FJ20 | 47.022 | 131.966 | 80 | 9.9 | 581 | 7 | 3.3 | 56.5 | -11.7 | -89.5 | nd |
| FJ21 | 47.013 | 132.143 | 100 | 8.9 | 379 | 7.42 | 3.3 | 55.3 | -11 | -84.6 | nd |
| FJ22 | 47.013 | 132.137 | 13 | 10 | 862 | 6.57 | 247 | 115 | -11.4 | -83.3 | nd |
| FJ23 | 47.021 | 132.179 | 80 | 10.3 | 493 | 7.26 | 3.7 | 54.5 | -11.4 | -87.4 | nd |
| FJ24 | 47.019 | 132.183 | 20 | 8.4 | 882 | 6.47 | 123 | 128 | -11.5 | -85.5 | nd |
| FJ25 | 46.998 | 132.263 | 80 | 8.9 | 387 | 7.21 | 3 | 41.6 | -12.1 | -90.8 | nd |
| FJ26 | 46.998 | 132.263 | 14 | 9 | 739 | 6.4 | 229 | 97.5 | -11.2 | -82.2 | nd |
| FJ27 | 46.955 | 132.46 | 19 | 15.6 | 390 | 6.87 | 1.3 | 51.8 | -11.9 | -86.5 | nd |
| FJ28 | 46.972 | 132.557 | 29 | 5.2 | 562 | 7.36 | 1.1 | 94.6 | -12.4 | -89.8 | nd |
| Groundwater-District II-confined | | | | | | | | | | | |
| DX01 | 46.937 | 132.891 | 20 | 7.7 | 249 | 7.03 | 2.6 | 29.9 | -12.2 | -89.7 | nd |
| DX02 | 46.892 | 132.902 | 58 | 7.2 | 441 | 7.21 | 3.6 | 48 | -12.3 | -92 | nd |
| DX03 | 46.893 | 132.9 | 23 | 8.3 | 487 | 6.7 | 1.3 | 49.7 | -12 | -88.4 | nd |
| DX04 | 46.974 | 132.968 | 20 | 10.2 | 216 | 6.75 | 1.9 | 21.9 | -12.1 | -88.6 | nd |
| DX05 | 46.997 | 133.071 | 23 | 9.1 | 282 | 6.56 | 6.4 | 28.7 | -12.3 | -90.7 | nd |
| DX06 | 46.993 | 133.094 | 23 | 3.3 | 313 | 6.7 | 1.2 | 31.9 | -12.3 | -90.3 | nd |
| DX07 | 47.049 | 133.165 | 40 | 8.6 | 285 | 6.7 | 7 | 25.9 | -11.6 | -86.6 | nd |
| DX09 | 47.019 | 133.291 | 10 | 2.9 | 997 | 7.39 | 84.5 | 104 | -11.9 | -88.9 | nd |
| DX10 | 47.013 | 133.291 | 103 | 9.1 | 260 | 6.95 | <0.05 | 16.6 | -12.6 | -91.6 | nd |
| DX11 | 47.049 | 133.078 | 9 | 9 | 516 | 6.4 | 84.9 | 44.2 | -11.4 | -83.7 | nd |
| DX12 | 46.975 | 132.88 | 120 | 9.1 | 532 | 6.94 | 5.4 | 67.2 | -11.9 | -89.7 | nd |
| DX13 | 46.979 | 132.777 | 40 | 8.1 | 248 | 6.93 | 2.9 | 32 | -12 | -88.1 | nd |
| DX14 | 46.983 | 132.71 | 30 | 7.9 | 187.8 | 6.85 | 1.8 | 56.5 | -9.9 | -78.6 | nd |
| DX15 | 46.979 | 132.654 | 20 | 7.2 | 275 | 6.95 | 1.4 | 32.4 | -11.7 | -86.6 | nd |
| DX16 | 46.978 | 132.65 | 80 | 7.6 | 916 | 6.96 | 7.4 | 121 | -8.7 | -73.5 | nd |
| HH1 | 47.513 | 133.511 | 40 | 6.4 | 92.7 | 6 | <0.05 | 21.9 | -12.8 | -96.8 | <1.0 |
| HH2 | 47.589 | 133.51 | 120 | 12.1 | 106.1 | 7 | <0.05 | 31.9 | -12.2 | -94.7 | <1.0 |
| HH3 | 47.69 | 133.482 | 15 | 5.9 | 69.1 | 7 | 1.7 | 16.2 | -12.6 | -92.9 | nd |
| HH4 | 47.515 | 133.513 | 30 | 8 | 75.2 | 6 | 1.7 | 17.3 | -12.9 | -95.2 | 1.5±1.3 |
| HH5 | 47.632 | 133.496 | 18 | 10.6 | 120.5 | 7 | 2.8 | 23.9 | -13.5 | -97.8 | nd |
| HH6 | 47.623 | 133.47 | 18 | 8.8 | 103.1 | 7 | 3.6 | 22.7 | -13.3 | -97.8 | nd |
| HH7 | 47.69 | 133.477 | 30 | 7.7 | 88.7 | 7 | 4.8 | 25.5 | -12.8 | -96.7 | <1.0 |
| HH8 | 47.589 | 133.501 | 18 | 11.7 | 52.7 | 7 | 5.4 | 11.1 | -12.4 | -92.6 | nd |
| HH9 | 47.586 | 133.502 | 18 | 11.2 | 100.7 | 7 | 6.6 | 27.4 | -13.2 | -98.6 | 1.9±1.3 |
| QF1 | 47.593 | 133.95 | 40 | 7.4 | 97.3 | 6 | <0.05 | 25.1 | -12.8 | -96.7 | nd |





| | | | | | | | | | | | |
|---|---|---|---|---|---|---|---|---|---|---|---|
| QF2 | 47.587 | 133.949 | 40 | 7.5 | 110.3 | 6 | <0.05 | 27.4 | -12.9 | -97.7 | nd |
| QF3 | 47.635 | 133.989 | 40 | 11.4 | 117.6 | 6 | 43.2 | 30.5 | -13.1 | -97 | nd |
| QF4 | 47.603 | 133.986 | 35 | 6.8 | 96.2 | 6 | <0.05 | 26.9 | -12.7 | -97 | <1.0 |
| QF5 | 47.597 | 133.885 | 60 | 9.6 | 117.8 | 7 | <0.05 | 31 | -12.3 | -94.7 | 2.2±1.3 |
| QF6 | 47.617 | 133.986 | 18 | 7.2 | 83.9 | 6 | 1.2 | 16 | -13.7 | -102.2 | nd |
| QF7 | 47.603 | 133.986 | 18 | 6.4 | 65.8 | 6 | 2.2 | 12.5 | -13.8 | -102.4 | <1.0 |
| QF8 | 47.633 | 133.987 | 48 | 10.1 | 88.8 | 6 | 4 | 21.4 | -13 | -99.2 | nd |
| QF9 | 47.559 | 133.952 | 30 | 7.4 | 109.8 | 6 | <0.05 | 39.6 | -10.8 | -86.1 | <1.0 |
| Groundwater-District III-unconfined | | | | | | | | | | | |
| QS1 | 48.026 | 134.166 | 20 | 8.7 | 36.3 | 5 | <0.05 | 8.8 | -10.3 | -81.7 | 53.0±2.7 |
| QS2 | 47.997 | 134.128 | 72 | 11.6 | 44.8 | 6 | 0.2 | 10.2 | -11.6 | -89.3 | 29.9±2.2 |
| QS3 | 47.982 | 134.149 | 56 | 8 | 58.5 | 6 | 1 | 13.8 | -13 | -98.5 | 6.5±1.8 |
| QS4 | 47.982 | 134.149 | 25 | 9.1 | 45.6 | 6 | 1.3 | 10.4 | -13.1 | -97.9 | nd |
| QS5 | 47.92 | 134.146 | 48 | 6.7 | 67 | 6 | 2.8 | 17.5 | -12.8 | -98.4 | nd |
| QS6 | 47.837 | 134.163 | 65 | 8.8 | 84.9 | 6 | 3.2 | 17.5 | -13.1 | -99.9 | <1.0 |
| QS7 | 47.948 | 134.144 | 24 | 7.8 | 77.5 | 6 | 4.2 | 16.7 | -12.9 | -98 | nd |
| QS8 | 47.874 | 134.155 | 45 | 6.9 | 72.4 | 6 | 5.5 | 16.9 | -13 | -99.2 | <1.0 |
| QS9 | 48.013 | 134.16 | 70 | 8.4 | 79.1 | 6 | 7.4 | 14.1 | -10.6 | -84.5 | <1.0 |
| QS10 | 48.012 | 134.153 | 20 | 8.1 | 68.9 | 6 | 10.1 | 17.4 | -12.5 | -92.7 | 71.3±2.7 |
| QS11 | 47.93 | 134.147 | 25 | 8.4 | 55.8 | 4 | 17.9 | 13.1 | -10.8 | -84.7 | 19.1±1.8 |
| Surface water | | | | | | | | | | | |
| S1 | 47.339 | 130.108 | river | 19 | 60.8 | 6.93 | 0.8 | 6.9 | -11.3 | -85 | nd |
| S2 | 47.203 | 130.817 | river | 23.7 | 194.5 | 7.09 | 2.7 | 20.6 | -10.2 | -80.4 | nd |
| S3 | 47.057 | 133.233 | river | 15.5 | 83.5 | 7.22 | 1.8 | 16.2 | -9.6 | -77.9 | nd |
| S4 | 48.014 | 134.153 | river | 23.8 | 37.2 | 4 | 1.5 | 6.8 | -10.1 | -74.4 | 25.1±2.0 |
| HH-P1 | 47.632 | 133.496 | Paddy | 22 | 33.3 | 5.5 | 1.9 | 6.5 | -10.6 | -74.5 | nd |
| HH-P2 | 47.623 | 133.47 | Paddy | 24.8 | 46.8 | 6.5 | 0.6 | 9.2 | -10.8 | -77 | nd |
| HH-P3 | 47.691 | 133.478 | Paddy | 30.3 | 40.3 | 6 | 3.3 | 6.3 | -10.2 | -72.9 | nd |
| HH-P4 | 47.69 | 133.477 | Paddy | 18.2 | 106.4 | 7 | 6 | 23.8 | -12.6 | -94.1 | nd |
| QF-P1 | 47.593 | 133.95 | Paddy | 21.5 | 78.8 | 6 | 1.2 | 9.8 | -11.1 | -84.7 | nd |
| QF-P2 | 47.587 | 133.949 | Paddy | 22.5 | 59.2 | 6 | 2.1 | 11.8 | -10.8 | -77.5 | nd |
| QF-P3 | 47.559 | 133.952 | Paddy | 21.8 | 32.9 | 5 | 1.3 | 6.1 | -10 | -71.7 | nd |
| QF-P4 | 47.617 | 133.986 | Paddy | 22.1 | 64.9 | 6 | <0.05 | 9.2 | -10.3 | -77.6 | nd |
| C1 | 47.69 | 133.482 | Channel | 32.1 | 66.3 | 6 | 3.7 | 7.2 | -9 | -73.1 | nd |
| C2 | 48.026 | 134.165 | Channel | 29.2 | 78.8 | 4 | 1.3 | 10.8 | -9 | -77 | nd |
| wetlands | 47.586 | 133.501 | wetlands | 24.8 | 114 | 4 | <0.05 | 3.4 | -11.7 | -85.9 | 22.4±2.0 |

1 Note:
2 [1]  Samples HH1 to HH10, QF1 to QF9, and QS1 to QS11 were collected from the Honghe, Qianfeng and Qianshao farms,
3    respectively.
4 [2]  Samples HG01 to DX16 were collected from the transaction A-A', from Hegang in the west to Daxing in the East.
5 [3]  "nd": no data





1     **Table 2 Stable isotopes in precipitation of the Sanjiang Observatory Station located in Honghe Farm**

| Date | P (mm) | $\delta^2H$ (‰) | $\delta^{18}O$ (‰) |
|---|---|---|---|
| 2005-01 | / | -207.3 | -28.2 |
| 2005-02 | 0.8 | -158.7 | -19.9 |
| 2005-03 | 4.0 | -100.9 | -13.4 |
| 2005-04 | 71.4 | -102.0 | -15.0 |
| 2005-05 | 60.0 | -92.4 | -13.1 |
| 2005-06 | 16.6 | -53.7 | -5.9 |
| 2005-07 | 129.4 | -61.2 | -8.2 |
| 2005-08 | 80.6 | -78.6 | -10.3 |
| 2005-09 | 51.0 | -67.6 | -8.8 |
| 2005-10 | 3.6 | -58.5 | -8.3 |
| 2005-11 | 3.8 | -125.1 | -14.8 |
| 2005-12 | 2.0 | -181.8 | -24.2 |
| 2006-01 | / | -54.5 | -7.6 |
| 2006-03 | 2.6 | -58.1 | -7.7 |
| 2006-04 | 32.4 | -141.5 | -18.8 |
| 2006-05 | 55.6 | -148.5 | -19.2 |
| 2006-06 | 111.4 | -38.26 | -4.74 |
| 2006-10 | 28.0 | -54.2 | -7.3 |
| 2006-11 | 21.2 | -75.2 | -8.3 |
| 2010-08 | 150.5 | -65.9 | -9.0 |
| 2010-09 | 31.4 | -68.8 | -9.3 |
| 2010-10 | 9.9 | -113.3 | -15.3 |
| 2010-11 | 27.4 | -178.5 | -24.2 |
| 2010-12 | 60.5 | -184.3 | -24.8 |
| 2011-01 | 11.0 | -180.0 | -22.9 |
| 2011-02 | 1.8 | -186.1 | -23.7 |
| 2011-03 | 9.9 | -169.4 | -22.7 |
| 2011-04 | 24.4 | -160.9 | -21.1 |
| 2011-05 | 42.9 | -94.5 | -12.9 |
| 2011-06 | 67.4 | -73.1 | -9.7 |
| 2011-07 | 56.2 | -86.4 | -11.7 |
| 2011-08 | 151.2 | -63.0 | -8.6 |
| 2011-09 | / | -106.6 | -14.4 |




1   **Table 3 Groundwater $^{14}$C content and corrected ages**

| District | Sample | $^{14}$C (pmC) | $\delta^{13}C_{DIC}$ (PDB) | Well Depth (m) | Age (year) | $^{3}$H |
|---|---|---|---|---|---|---|
| I and III | FJ14 | 30.7 | -13.7 | 152 | 7910 | |
| | FJ02 | 72.6 | -15.8 | 60 | 800 | |
| | HC03 | 76.6 | -14.2 | 100 | 360 | |
| | HG10 | 75.1 | -17.3 | 100 | 520 | |
| | HG01 | 69.3 | -16.7 | 123 | 1190 | |
| | QS1 | 98.4 | -20.1 | 20 | modern | 29.9 |
| II | DX12 | 67.4 | -13.5 | 120 | 1420 | |
| | DX02 | 65.1 | -13.6 | 58 | 1700 | |
| | HH2 | 70.3 | -18.0 | 120 | 1070 | <1.0 |
| | HH1 | 62.4 | -16.3 | 40 | 2060 | <1.0 |
| | QF1 | 73.6 | -15.6 | 40 | 690 | 2.2 |

