# Peer review of "Manuscript under review for journal Hydrol. Earth Syst. Sci."

_Hydrology and Earth System Sciences, 2016_

## Referee Comment (RC1) · M. Currell (Referee) · 16 Jun 2016

General comments: Pang et al. present a manuscript describing their study which employs isotopic tracers to examine the prospects for sustainable groundwater usage in agriculture in an area of reclaimed wetland in northeast China.

Overall, the methods adopted here are not particularly novel (many other studies have been published using similar tracers), but the study is potentially of international significance, given the importance of groundwater sustainability in agriculture in northern China - a region of globally significant agriculture facing a major water crisis. The sampling and analytical campaign is well-designed and the data-sets are of good quality. The paper is generally well written (although there are sections that need some improvement), and the figures clear and informative.

[Figure]

However, there are deficiencies which I believe need to be addressed prior to publication. In particular, more rigour is needed in the processing and interpretation of the isotopic tracer data. Limitations of the methods used to arrive at estimates of groundwater age and categorisation of water into 'modern' and 'pre-modern' need to be acknowledged and discussed in much more detail. The links between groundwater and surface water also need to be more clearly demonstrated with reference to the data and figures (such as maps and spatially referenced comparisons between surface water and groundwater levels). The discussion is also too brief and lacks depth and detail at this stage. The link between groundwater age and recharge mechanism, and groundwater sustainability is not explained clearly enough. Are the authors proposing that low recharge rates and a lack of tritium indicate 'pre-modern' water in the confined aquifer, and thus that there is a limit to the sustainable extraction rate from this system? If so, this should be carefully explained and the potential for 'capture' of water from other areas (and release of water from aquitards) explored. There may be water quality implications for high rates of groundwater extraction also, as documented in Currell et al. Journal of Hydrology 385 pp 216-225. With regard to the unconfined aquifer, it appears that there is active recharge, on the basis of high nitrate and tritium concentrations observed in shallow groundwater. Is this attributed to recharge through irrigation return leakage, rainfall recharge, surface water leakage, or some combination of the three? Is groundwater quality a limiting factor for the utilisation of the unconfined aquifer groundwater (e.g. because of the high nitrate concentrations)? These issues should all be clearly explained with reference to the data and more detailed discussion of the trends observed in different parts of the study area. I think some further figures such as maps showing the distribution of tritium and perhaps nitrate in the aquifers will be illustrative of the areas where recharge is actively occurring.

If these issues (and the specific comments below) can be addressed, along with some required technical corrections, I believe the paper may be suitable for publication.

Specific comments: The editor has noted that the authors need to provide background

on the purpose of each analysis and more detail about the stable isotope evaporation model and tritium decay model. These areas have been addressed to some degree. However, I question how robust the use of the Ottawa tritium in precipitation record is for the study area, given there have been only 4 samples collected at the local IAEA station for comparison. The 'latitude effect' is not clearly explained; do you mean that because the two stations are at similar latitude we can infer the tritium records are expected to be approximately the same? Some explanation and one or more references for this assumption is needed here. It should also be made clear that the age estimation using tritium is only a semi-quantitative tool (as the 3H-He method is not adopted).

Abstract The abstract needs some more reference to the data and more context. e.g. Is groundwater quality the limiting factor for groundwater utilisation from the unconfined aquifer? If so, then what particular aspects of water quality are important? What is the link between groundwater age and recharge mechanism, and sustainability of groundwater usage? For example, groundwater extraction from the confined aquifers will induce flow and leakage from other areas, is the quality of the induced flow a potential limiting factor (as in other areas in China)?

Introduction The introduction and background information are concise and generally informative.

Methods Further information is needed on the sample collection methods for groundwater and surface water. Are the groundwater samples from production wells, or monitoring wells? What is the range of sample depths and screened intervals? For surface water, were the samples 'grab samples'? If so, at what time of year were they taken? This may impact whether the samples represent recent runoff, snow melt and/or water impacted by evaporative enrichment.

The LMWL should be calculated using a weighted regression method, as described in Hughes and Crawford, Journal of Hydrology 464-465 pp 344-351 (2012), rather than simple linear regression.

Results The relationship between lithology and ion composition (e.g. Ca) should be discussed and examined in more detail. Are carbonate minerals in the soil and/or aquifer the likely source of Ca? Is fertiliser a potential source also? A plot of the Ca vs 13C isotopes would be helpful in this context. You may also consider including and discussing the full dataset on water major ion chemistry, and discuss TDS distribution in the aquifers.

Plotting tritium and carbon-14 data vs sample depth would be useful, and also plotting tritium concentrations on a map. This would allow better assessment of where spatially the recent and 'pre-modern' water samples are distributed with respect to current agricultural irrigation areas, and it will help to better identify areas of 'active recharge' as distinct from those not receiving such recharge.

Discussion The writing in the discussion needs some further improvement; technical corrections are suggested below but these are not exhaustive.

-When discussing 'vertical infiltration' as a recharge mechanism (e.g. p. 10) you should distinguish between recharge due to rainfall infiltration and/or irrigation return-flow, and recharge from surface water bodies such as rivers. -The use of the tritium/radiocarbon plot to estimate initial activities of 14C has some merit, however it should be conducted more rigorously, explained in more detail, and used with some caution. Are you using a linear extrapolation between 'modern' and tritium free water in the various samples to arrive at the initial pMC of approximately 80? What about the influence of mixing between 'young' water and older water (which should produce a straight line relationship, as opposed to a decay-based curve)? Decay and mixing will produce different patterns in 3H and 14C and this needs to be carefully analysed. For further detail refer to Cartwright et al, Journal of Hydrology 380 pp. 203-221 (2010), particularly Figure 8. The use of this method does not discount the need to assess other potential sources of DIC and influences on initial 14C activities. A plot of the 13C vs 14C data is needed, as is some further analysis of the ion chemistry (e.g. Ca vs 13C) to shore up this area. -As indicated above, the link between groundwater age, recharge and groundwater sustainability is not explained clearly enough. You need to put more work into defining (on your maps) where groundwater is influenced by direct vertical recharge, river recharge and lateral recharge, and discuss the water quality implications of these different mechanisms. Where in particular do you think the extraction rates for groundwater are likely to be much greater than recharge? What is the likely response of the aquifer(s) to extraction and is there any water level data to show what is happening currently? What are the likely water quality implications of extraction from different aquifers and depths (see previous comments)? -Overall the discussion is too brief, and further discussion of limitations of your isotopic data, and alternative explanations need to be explored and discounted.

Technical corrections: p2 Line 10 'Recharge and regime', do you mean 'recharge and groundwater flow patterns'? Lines 11 & 12: Grammar is poor. Do you mean 'with ages over 50y is recharged by lateral flow..as evidenced by depleted heavy isotopes'? Which isotopes (I assume 18O and 2H)? Line 27: Citation (Assessment, 2005) is incorrect. A suggested citation format is given in the front matter of this report. P4. Line 22: Suggest using ML rather than mega-L P5. Line 2: 'hydrogeology' should be 'hydrogeological' P6. Line 19-20: Can remove the statement 'our current efforts...tracers'. It is better to clearly outline your study aims and scope in the introduction section

---

## Referee Comment (RC2) · Anonymous Referee #2 · 12 Jul 2016

The paper is a case study of the agriculturally used Sanjiang plain. While the paper presents some interesting data I do no support publication for a number of reasons. The paper presents a very large amount of data, but the goal of this research remains unclear until the end. The introduction highlights that the goal of the paper is to understand "the implications of sustainable irrigation agriculture" (implications on what? Sustainable in terms of what?) as well as the "factors controlling groundwater regime". These goals are fairly unspecific, and I do not think they have been achieved in the study. For example, stable isotpoes are widely used in the section recharge to aquifers, but there is not a single estimate of recharge rates. The same is true for the residence times. To be publishable, I suggest that the research questions are much more specific, and that the data are used for a quantitative interpretation. Right now the paper reads like a long and somewhat random collection of data without too much

quantitative substance or research context.

---

## Referee Comment (RC3) · Anonymous Referee #3 · 26 Jul 2016

Summary:

Pang et Al. use several tracers (major ions, water isotopes, carbon isotopes) trying to determine wether agricultural irrigation using groundwater can be considered sustainable. This is done by analyzing the interplay of groundwater, surface waters and agricultural practices in a wetland region in north-eastern China.

General Comments:

While the presented data set seems to be very interesting and may provide new information on ecological impacts of groundwater based irrigation practices the general techniques have been published by other studies before. The paper is written well (from my point of view as a non-native speaker), but the method section misses detailed information on the analytical procedures (as already asked for by the editor) and

there are some unclear conclusions. At the present state of this paper, I do not support publication in HESS for several reasons: a) the research question is very unspecific, i.e. there is no clear hypothesis which allows the reader to understand the underlying plan of the study-design, b) some of the conclusions seem to be drawn more by guessing than by quantitative analysis and c) the question stated in the manuscript title is not answered in a quantitative way by the analysis presented in the paper. Nevertheless, I think that with some restructuring, a specification of the article-focus and some efforts the study might be interesting for the readership and thus publishable in HESS.

Specific Comments:

Introduction

The introduction section misses a brief introduction on the previous knowledge about the interplay between irrigation practices and recharge mechanisms for confined and unconfined aquifers with a clear statement of the research gaps which will be closed by this paper.

Study Area

While the results and the discussion are presented with respect to particular sampling locations there is no spatial information on the locations of these sampling locations (I have seen the coordinates in the tables, but this doesn't help/ would take a lot of time to locate the different stations on the map).

Methods

There is no clear methodology/procedure which explains how the results of the chemical analysis are treated. This also marks the big lack in this paper: There is no quantitative analysis of observed concentrations. For example, the presented nitrate and calcium concentrations are only "analyzed" with a rather surficial interpretation of "concentration groups" which does not fit at all (see comments on the result section). The method section would need a clear concept how the results of the chemical analysis

were sorted, ranked, correlated, . . . and a hypothesis how this procedure will lead to the answers sought by this paper.

Results

In general, the results miss any quantitative information on how ". . .groundwater is sufficient to support sustainable irrigation agriculture in a reclaimed wetland region". While the results of the isotopic analysis show the overlay of surface and groundwaters the interpretation of the major ion concentrations does not fit at all: Figures 3 and 4 show similar concentrations for Districts II and III either for nitrate and for calcium and not as presented for District I and III. Consequently, the following interpretations should be reassessed. The Deuterium enrichment in the paddy field water samples is interpreted as condensation (Figure 6). This is wrong, condensation fractionates along the LMWL (saturated conditions). A possible reason might be methanogenesis which can cause heavy Deuterium-enrichment of soil water.

E. g. Chidthaisong, A., Chin, K. J., Valentine, D. L., & Tyler, S. C. (2002). A comparison of isotope frac-tionation of carbon and hydrogen from paddy field rice roots and soil bacterial enrichments during $CO_2/H_2$ methanogenesis. Geochimica et Cosmochimica Acta, 66(6), 983-995.

Discussion

The discussion section misses for the largest parts the reflection of the actual literature with the results and the determination how the results presented within this study contribute to our understanding of the governing processes. For example, there is no explanation how the results of the groundwater age dating correspond to the various major ion concentrations (e.g. nitrate) for the different aquifer types and which recharge processes could cause observable chemical groundwater compositions.

---

## Author Comment (AC1) · 26 Aug 2016

Thank you, Dr. Currell, for your comments concerning our manuscript entitled "Is groundwater sufficient to support sustainable irrigation agriculture in a reclaimed wetland region?" (MS No. hess-2016-155). Your comments are very valuable and helpful for improving our manuscript. We have numbered your comments for clarity. Our responses are described one by one in the following:

Comment 1: Limitations of the methods used to arrive at estimates of groundwater age and categorisation of water into 'modern' and 'pre-modern' need to be acknowledged and discussed in much more detail.

Reply 1: You are right. Due to the fact that tritium in precipitation in most parts of the world has returned to natural background, it is becoming more and more difficult to use

it as an age dating tool for groundwater. In this study, we have used the 3H content versus 14C activity plot to determine the initial 14C activity. It gives a good indication of the initial 14C activity. The 3H contents of groundwater in the confined aquifers are very low (<2.2TU) and the 14C ages are thousands of years, which indicates that the groundwater in the confined aquifers contains no modern component, referred to as pre-modern water. However, groundwater in the unconfined aquifers shows a wide range of 3H values. Groundwater samples near the river contain higher levels of 3H (6.5-71.3TU), indicating the existence of modern water component.

Comment 2: The links between groundwater and surface water also need to be more clearly demonstrated with reference to the data and figures (such as maps and spatially referenced comparisons between surface water and groundwater levels).

Reply 2: There are strong evidences to support the existence of linkages between groundwater and surface water. Isotopic composition of groundwater in the unconfined aquifers is more enriched than that of groundwater in the confined aquifers and ground-water samples in the unconfined aquifers are plotted on the local evaporation line (Fig. 7 and Fig. 8). High NO3- and Ca2+ concentrations can be found in the shallow un-confined groundwater (Districts I and III) (Fig. 3 and Fig. 4), indicating the interactions with irrigation water, while those in the confined area are generally low. Groundwater in the unconfined aquifers shows a wide range of 3H values. Groundwater samples near the river contain high levels of 3H (6.5-71.3TU) indicating their linkage with river water. In a word, the groundwater in the unconfined aquifer has strong links with the surface water, while such links in the confined aquifers are very weak, if any, because there is a thick clay layer on top of it that separates them.

Comment 3: The link between groundwater age and recharge mechanism, and ground-water sustainability is not explained clearly enough. Are the authors proposing that low recharge rates and a lack of tritium indicate 'pre-modern' water in the confined aquifer, and thus that there is a limit to the sustainable extraction rate from this system? If so, this should be carefully explained and the potential for 'capture' of water from other

areas (and release of water from aquitards) explored.

Reply 3: The low 3H values in the confined area indicate that the groundwater is pre-modern and high 14C ages show that the recharge rate is very low. The fast decline of water table is also an indication that the aquifers get very limited recharge and potential other sources, e.g. release of aquitards, is also limited. It is therefore justifiable to conclude that groundwater alone is not sufficient to support sustainable irrigation agriculture. Integrated use of groundwater and surface water is a better solution to sustain irrigation agriculture in such regions.

Comment 4: There may be water quality implications for high rates of groundwater extraction also, as documented in Currell et al. Journal of Hydrology 385 pp 216-225.

Reply 4: We agree with you. As shown in Fig. 8, the water table in the confined aquifers shows a trend of gradually declining while that in the unconfined aquifers remains relatively stable. Some samples with high NO3- and Ca2+ concentrations (especially NO3-) in the confined area may indicate leakage from the unconfined groundwater similar to that reported in Currell et al., 2010. If the pumping in the confined area continues at or increases from the present levels, the groundwater table declining will continue and the water quality will deteriorate, which is indeed unsustainable.

Comment 5: With regard to the unconfined aquifer, it appears that there is active recharge, on the basis of high nitrate and tritium concentrations observed in shallow groundwater. Is this attributed to recharge through irrigation return leakage, rainfall recharge, surface water leakage, or some combination of the three? Is groundwater quality a limiting factor for the utilisation of the unconfined aquifer groundwater (e.g. because of the high nitrate concentrations)? These issues should all be clearly explained with reference to the data and more detailed discussion of the trends observed in different parts of the study area. I think some further figures such as maps showing the distribution of tritium and perhaps nitrate in the aquifers will be illustrative of the areas where recharge is actively occurring.
Reply 5: Yes, there is a need to add more discussion. Based on the enriched 18O and 2H isotopic compositions, high NO3- and Ca2+ concentrations (especially NO3-) (Fig. 3 and Fig. 4) and high 3H values near the river in the unconfined aquifers, we believe that the combination of irrigation return leakage, rainfall and surface water leakage have all contributed to the active recharge in the unconfined aquifers. On the basis of high NO3- concentrations, groundwater quality is indeed a limiting factor in the long term for the utilization of the unconfined groundwater. Furthermore, groundwater quality will also be a limiting factor for the sustainable utilization of it in the confined area as shallow groundwater leakage will occur in the future if the pumping in the confined area continues at or increases from the present levels.

Comment 6: The editor has noted that the authors need to provide background on the purpose of each analysis and more detail about the stable isotope evaporation model and tritium decay model. These areas have been addressed to some degree. However, I question how robust the use of the Ottawa tritium in precipitation record is for the study area, given there have been only 4 samples collected at the local IAEA station for comparison.

Reply 6: Atmospheric circulation in the stratosphere is controlled by latitude, resulting in the zonal distribution of tritium ("latitude effect" in the manuscript) in precipitation (Eriksson, E.: An account of the major pulses of tritium and their effects in the atmosphere, Tellus, 17, 118-130, doi: 10.1111/j.2153-3490.1965.tb00201.x, 1965; Clark and Fritz, 1997). On the other hand, tritium levels recorded near the study area, at the Qiqihar GNIP station (latitude 47°23'0"), show a similar trend to that in Ottawa, Canada (latitude 45°23'0"), during an overlapping period, and the two stations are at similar latitude. So we infer that the precipitation tritium record of Ottawa can be used to approximate the input function of tritium in precipitation in the Sanjiang Plain.

Comment 7: The 'latitude effect' is not clearly explained; do you mean that because the two stations are at similar latitude we can infer the tritium records are expected to be approximately the same? Some explanation and one or more references for this
assumption is needed here.

Reply 7: See "Reply 6".

Comment 8: It should also be made clear that the age estimation using tritium is only a semi-quantitative tool (as the 3H-He method is not adopted).

Reply 8: We agree with you. The age estimation using tritium is only a semi-quantitative tool. We wanted to determine if there is modern water component in the groundwater of confined or unconfined aquifers. It seems sufficiently accurate to serve our purposes in this study.

Comment 9: The abstract needs some more reference to the data and more context. e.g. Is groundwater quality the limiting factor for groundwater utilisation from the unconfined aquifer? If so, then what particular aspects of water quality are important? What is the link between groundwater age and recharge mechanism, and sustainability of groundwater usage? For example, groundwater extraction from the confined aquifers will induce flow and leakage from other areas, is the quality of the induced flow a potential limiting factor (as in other areas in China)?

Reply 9: Yes, we can add more context to the abstract. Although active recharge occurs in the unconfined aquifers and the water table is relatively stable, the water quality with relatively high NO3- concentrations is a limiting factor for sustainable groundwater utilization. With weak interactions with surface water and lateral flow as the main recharge source, the groundwater age is older in the confined aquifers. The continuously declining water table demonstrates that groundwater is not sufficient for sustainable irrigation agriculture in terms of water quantity, though it is not severe at present. Furthermore, groundwater quality will also be a limiting factor for the sustainable utilization of groundwater in the confined aquifers as shallow groundwater leakage will occur in the future if the pumping in the confined aquifers continues at the present or increased levels.

Comment 10: Methods Further information is needed on the sample collection methods for groundwater and surface water. Are the groundwater samples from production wells, or monitoring wells? What is the range of sample depths and screened intervals? For surface water, were the samples 'grab samples'? If so, at what time of year were they taken? This may impact whether the samples represent recent runoff, snow melt and/or water impacted by evaporative enrichment.

Reply 10: We will add the information about water samples collection. The groundwater samples are from production wells. The sample depths are presented in Table 1 and groundwater samples are the mixing of water within the well depth intervals. The surface water samples S1, S2, S3 were taken at 2011 from rivers along the transect A-A' in Fig. 1. The other surface water samples were taken at 2009 from paddy fields, drainage channels and rivers within farms HH, QF and QS in the northeast part of the Sanjiang Plain (Fig. 1).

Comment 11: The LMWL should be calculated using a weighted regression method, as described in Hughes and Crawford, Journal of Hydrology 464-465 pp 344-351 (2012), rather than simple linear regression.

Reply 11: We have calculated using the PWLSR method as described in Hughes and Crawford, Journal of Hydrology, 464-465, 344-351 (2012), and the LMWL is $y=(7.39\pm0.14)x-0.88$ (R2=0.99), which is very similar to "y=7.51x-0.92" in the manuscript. We will replace it.

Comment 12: Results The relationship between lithology and ion composition (e.g. Ca) should be discussed and examined in more detail. Are carbonate minerals in the soil and/or aquifer the likely source of Ca? Is fertiliser a potential source also? A plot of the Ca vs 13C isotopes would be helpful in this context. You may also consider including and discussing the full dataset on water major ion chemistry, and discuss TDS distribution in the aquifers.

Reply 12: We think a plot of 13C vs 14C is a good indication of carbonate minerals dissolution. As shown in Fig. 10, there is no significant negative correlation relationship between 13C and 14C, indicating carbonate minerals dissolution is not the main source of Ca in the aquifers. However, Ca contents of the soil in Sanjiang Plain are high. The unconfined aquifer is influenced by vertical infiltration, and the interaction with the calcium-rich soil leads to the high calcium concentrations in the shallow groundwater. In the confined aquifer, lateral flow dominates the groundwater recharge. Lack of carbonate in the confined aquifers results in the relatively low Ca contents. Fertilizer is not a potential source of Ca as no calcium fertilizer has been used.

Comment 13: Plotting tritium and carbon-14 data vs sample depth would be useful, and also plotting tritium concentrations on a map. This would allow better assessment of where spatially the recent and 'pre-modern' water samples are distributed with respect to current agricultural irrigation areas, and it will help to better identify areas of 'active recharge' as distinct from those not receiving such recharge.

Reply 13: We presented 3H and 14C distributions in different areas with distinct hydrogeological conditions. Groundwater at QS farm (in unconfined area) shows a wide range of tritium levels (<1.0-71.3TU), corresponding to the different sampling locations. Samples with high levels of tritium (6.5-71.3TU) are from shallow groundwater collected near rivers, and those collected away from the river are from deeper groundwater showing low levels of tritium (<1.0TU). On the other hand, groundwater samples in the unconfined area generally have enriched 2H and 18O isotopic compositions and high NO3- and Ca2+ concentrations. These indicate that groundwater in the unconfined area have strong links with surface water and active recharge occurred. However, in the confined area (Districts II), low tritium levels and NO3- and Ca2+ concentrations, depleted 2H and 18O isotopic compositions and high 14C age indicate groundwater has weak interactions with surface water without modern water recharge.

Comment 14: -When discussing 'vertical infiltration' as a recharge mechanism (e.g. p. 10) you should distinguish between recharge due to rainfall infiltration and/or irrigation return-flow, and recharge from surface water bodies such as rivers.

Reply 14: As discussed in the manuscript, the samples near the rivers have high 3H values indicating recharge from rivers. Those samples with high NO3- concentrations and enriched 2H and 18O isotopic compositions indicate vertical infiltration from rainfall and irrigation water.

Comment 15: -The use of the tritium/radiocarbon plot to estimate initial activities of 14C has some merit, however it should be conducted more rigorously, explained in more detail, and used with some caution. Are you using a linear extrapolation between 'modern' and tritium free water in the various samples to arrive at the initial pMC of approximately 80? What about the influence of mixing between 'young' water and older water (which should produce a straight line relationship, as opposed to a decay-based curve)? Decay and mixing will produce different patterns in 3H and 14C and this needs to be carefully analysed. For further detail refer to Cartwright et al, Journal of Hydrology 380 pp. 203-221 (2010), particularly Figure 8. The use of this method does not discount the need to assess other potential sources of DIC and influences on initial 14C activities. A plot of the 13C vs 14C data is needed, as is some further analysis of the ion chemistry (e.g. Ca vs 13C) to shore up this area.

Reply 15: We agree with you. As shown in Fig. 10, there is a decay-based curve relationship rather than straight line relationship between 14C and 3H, which is similar to that shown in Cartwright et al, 2010. So the mixing between modern water and older water is precluded. To arrive at the initial 14C activity, the decay-based curve relationship was used rather than straight line relationship between 14C and 3H. Based on the decay-based curve relationship between 14C and 3H, when 3H value is below the 3H detection limit, or in other words, the water is Tritium-free, the corresponding 14C activity can be considered to be the initial 14C value. Carbonate minerals dissolution should be considered when using 14C age. We think a plot of 13C vs 14C is a good indication of carbonate minerals dissolution. As shown in Fig. 10, there is no significant negative correlation relationship between 13C and 14C, indicating carbonate minerals dissolution does not dominate in the aquifers.

Comment 16: -As indicated above, the link between groundwater age, recharge and ground-water sustainability is not explained clearly enough. You need to put more work into defining (on your maps) where groundwater is influenced by direct vertical recharge, river recharge and lateral recharge, and discuss the water quality implications of these different mechanisms. Where in particular do you think the extraction rates for groundwater are likely to be much greater than recharge? What is the likely response of the aquifer(s) to extraction and is there any water level data to show what is happening currently? What are the likely water quality implications of extraction from different aquifers and depths (see previous comments)?

Reply 16: As is discussed in the manuscript, in the unconfined aquifers (Districts I and III), the samples near the rivers contain high 3H values, indicating recharge from rivers. Those samples with high NO3- concentrations and enriched 2H and 18O isotopic compositions indicate vertical infiltration from rainfall and irrigation water. Although active recharge occurred in the unconfined area and the groundwater table is relatively stable (Fig. 8), the water quality with high NO3- concentrations is a limiting factor for sustainable groundwater utilization. In the confined area (Districts II), with weak interactions with surface water and lateral flow as the main recharge source, the groundwater age is older. The continuous declining groundwater table (Fig. 8) demonstrated that groundwater is not sufficient for sustainable irrigation agriculture in terms of water quantity. Furthermore, groundwater quality will also be a limiting factor for the sustainable utilization of groundwater in the confined area as shallow groundwater leakage will occur in the future if the pumping in the confined area continues at the present or increased levels.

Comment 17: -Overall the discussion is too brief, and further discussion of limitations of your isotopic data, and alternative explanations need to be explored and discounted.

Reply 17: We will rewrite the discussion section with more details discussed above.

Comment 18: p2 Line 10 'Recharge and regime', do you mean 'recharge and groundwater flow patterns'?

Reply 18: We mean the groundwater recharge and the change of groundwater table.

Comment 19: Lines 11 & 12: Grammar is poor. Do you mean 'with ages over 50y is recharged by lateral flow.as evidenced by depleted heavy isotopes'? Which isotopes (I assume 18O and 2H)?

Reply 19: We will rewrite the sentence as follows: Groundwater in the confined Quaternary aquifer with ages over 50 years and evidenced by depleted 18O and 2H isotopic compositions is recharged by lateral flow from nearby mountains.

Comment 20: Line 27: Citation (Assessment, 2005) is incorrect. A suggested citation format is given in the front matter of this report.

Reply 20: We will rewrite it as follow: Millennium Ecosystem Assessment: Ecosystems and human well-being: wetlands and water, World Resources Institute, Washington, DC, 2005. The citation in the text is "(Millennium Ecosystem Assessment, 2005)".

Comment 21: P4. Line 22: Suggest using ML rather than mega-L.

Reply 21: We will use ML instead of mega L.

Comment 22: P5. Line 2: 'hydrogeology' should be 'hydrogeological'.

Reply 22: We will use hydrogeological instead of hydrogeology.

Comment 23: P6. Line 19-20: Can remove the statement 'our current efforts...tracers'. It is better to clearly outline your study aims and scope in the introduction section.

Reply 23: We agree with you and we will remove the statement from the "Sampling and analyses" section.

---

## Author Comment (AC2) · 26 Aug 2016

Thank you for your comments concerning our manuscript entitled "Is groundwater sufficient to support sustainable irrigation agriculture in a reclaimed wetland region?" (MS No.: hess-2016-155). Those comments are valuable and helpful for improving our manuscript. We have numbered the comments for clarity. Responses are described one by one as follows:

Comment 1: The paper presents a very large amount of data, but the goal of this research remains unclear until the end. The introduction highlights that the goal of the paper is to understand "the implications of sustainable irrigation agriculture" (implications on what? Sustainable in terms of what?) as well as the "factors controlling groundwater regime". These goals are fairly unspecific, and I do not think they have

been achieved in the study. For example, stable isotopes are widely used in the section recharge to aquifers, but there is not a single estimate of recharge rates. The same is true for the residence times.

Reply 1: Focused on understanding of groundwater residence times, recharge mechanisms and the interactions between groundwater and surface water, we tried to figure out whether groundwater is sufficient to support sustainable irrigation agriculture in terms of water quantity and quality. From the perspective of stable water isotopes, isotopic compositions of groundwater in the unconfined area are more enriched than that of groundwater in the confined area and some groundwater samples in the unconfined area is located on the local evaporation line (Fig. 7 and Fig. 8), indicating the links of the unconfined groundwater with the surface water. From the perspective of hydrochemistry, high $NO_3-$ and $Ca_2+$ concentrations can be found in the shallow groundwater in the unconfined area (Districts I and III), indicating the interactions with irrigation water, while those in the confined area are generally low. From the perspective of 3H values, groundwater in the unconfined area shows a wide range of 3H values. Especially those groundwater samples near the river have high levels of 3H (6.5-71.3TU) indicating the links with river water. In a word, the groundwater in the unconfined area have strong links with the surface water, while groundwater in the confined area largely recharged by lateral flow. We found that hydrogeological conditions are the main controlling factors. In District I and III of the study area, the aquifers are composed of highly permeable cobble and gravel deposits and unconfined. In contrast, in the eastern part (District II), the aquifer is covered by a 16-20m thick clay layer and confined or semi-confined. The low 3H values in the confined area indicate that the groundwater is pre-modern and high 14C ages show that the recharge rate is very low. Based on the continuous decline of groundwater table, the groundwater alone is not sufficient to support sustainable irrigation agriculture. In the unconfined area, while the groundwater has strong links with surface water and relatively high recharge rate with stable groundwater table, the water quality is deteriorating affected by surface water which is unsustainable for irrigation agriculture. Furthermore, some samples with high
[Figure]

NO3- and Ca2+ concentrations (especially NO3-) in the confined area may indicate the leakage from the shallow unconfined groundwater. If the pumping in the confined area continues at or increases from the present levels, the groundwater table declining will continue and the water quality will also deteriorate in the future, which is also indeed unsustainable.

Comment 2: To be publishable, I suggest that the research questions are much more specific, and that the data are used for a quantitative interpretation. Right now the paper reads like a long and somewhat random collection of data without too much quantitative substance or research context.

Reply 2: In this study, focused on understanding of groundwater residence times, recharge mechanisms and the interactions between groundwater and surface water in areas with different hydrogeological conditions, we tried to figure out whether ground-water is sufficient to support sustainable irrigation agriculture in terms of water quantity and quality. As a matter of fact, the fast decline of water table in the confined aquifers called for the research conducted. The study is sufficiently quantitative to answer the question of sustainability and if another source of water support if necessary.

---

## Author Comment (AC3) · 26 Aug 2016

Thank you for your comments concerning our manuscript entitled "Is groundwater sufficient to support sustainable irrigation agriculture in a reclaimed wetland region?" (MS No.: hess-2016-155). Those comments are valuable and helpful for improving our manuscript. We have numbered the comments for clarity. Responses are described one by one as follows:

Comment 1: a) the research question is very unspecific, i.e. there is no clear hypothesis which allows the reader to understand the underlying plan of the study-design.

Reply 1: In this study, we tried to figure out whether groundwater is sufficient to support sustainable irrigation agriculture in terms of water quantity and quality. We tried to answer the question from the perspectives of groundwater residence times, recharge

mechanisms, interactions with surface water and groundwater regime with evidences from hydrogeochemical and isotopic tracers. Following the first sampling program in 2009, we wanted to explore whether the groundwater systems behave differently with different hydrogeological settings, then the second sampling program for hydrogeochemistry and isotopes was conducted in 2011 along the transect A-A' across different hydrogeological settings. Finally, the hydrogeochemical and isotopic tracers data was analyzed and discussed to answer the question. From the perspective of stable water isotopes, isotopic compositions of groundwater in the unconfined area are more enriched than that of groundwater in the confined area and some groundwater samples in the unconfined area is located on the local evaporation line (Fig. 7 and Fig. 8), indicating the links of the unconfined groundwater with the surface water. From the perspective of hydrochemistry, high $NO_3^-$ and $Ca^{2+}$ concentrations can be found in the shallow groundwater in the unconfined area (Districts I and III), indicating the interactions with irrigation water, while those in the confined area are generally low. From the perspective of 3H values, groundwater in the unconfined area shows a wide range of 3H values. Especially those groundwater samples near the river have high levels of 3H (6.5-71.3TU) indicating the links with river water. In a word, the groundwater in the unconfined area have strong links with the surface water, while groundwater in the confined area largely recharged by lateral flow. We found that hydrogeological conditions are the main controlling factors. In District I and III of the study area, the aquifers are composed of highly permeable cobble and gravel deposits and unconfined. In contrast, in the eastern part (District II), the aquifer is covered by a 16-20m thick clay layer and confined or semi-confined. The low 3H values in the confined area indicate that the groundwater is pre-modern and high 14C ages show that the recharge rate is very low. Based on the continuous decline of groundwater table, the groundwater alone is not sufficient to support sustainable irrigation agriculture. In the unconfined area, while the groundwater has strong links with surface water and relatively high recharge rate with stable groundwater table, the water quality is deteriorating affected by surface water which is unsustainable for irrigation agriculture. Furthermore, some samples with high

NO3- and Ca2+ concentrations (especially NO3-) in the confined area may indicate the leakage from the shallow unconfined groundwater. If the pumping in the confined area continues at or increases from the present levels, the groundwater table declining will continue and the water quality will also deteriorate in the future, which is also indeed unsustainable.

Comment 2: the question stated in the manuscript title is not answered in a quantitative way by the analysis presented in the paper.

Reply 2: It is our conclusion that groundwater is insufficient to support irrigation agriculture, an answer to the question raised in the title of the paper. This answer has been supported by scientific evidences including groundwater residence time, recharge mechanisms and the interaction between groundwater and surface water in areas with different hydrogeological conditions, typically for a reclaimed wetlands.

Comment 3: Introduction The introduction section misses a brief introduction on the previous knowledge about the interplay between irrigation practices and recharge mechanisms for confined and unconfined aquifers with a clear statement of the research gaps which will be closed by this paper.

Reply 3: You are right. This can be further explained and literature cited. It is especially interesting to note that, groundwater is a vital source of drinking water and irrigation water in north China, especially in areas of former wetlands, such as Sanjiang Plain, the study area of this paper, people believe that groundwater is sufficient. However, problems have occurred in north China plain, an area with irrigation agriculture. Many shallow unconfined aquifers in north China have been contaminated by nitrate and other pollutants with recharge from surface water due to agricultural activities (Chen et al., Journal of Hydrology, 326, 367–378, 2006; Zhu et al., Hydrogeology Journal, 16, 167–182, 2008). Consequently, more and more deep confined groundwater has been used for irrigation agriculture and drinking water. The deep confined groundwater may not be replenished by modern recharge and the recharge rate is low (Edmunds

et al., 2006). Continuous exploitation of confined groundwater may cause water table decline and there is also the potential for deep groundwater that is not fully confined to be contaminated by downward leakage from overlying shallow groundwater. In this study, we apply a multi-tracer approach to demonstrate that groundwater may not get appropriate recharge so it is not sustainable to use it as a sole source. Environmental tracers have been demonstrated as useful tools in understanding groundwater residence times, recharge mechanisms and the interactions between groundwater and surface water in a wetland terrain with diverse hydrogeological settings.

Comment 4: Study Area While the results and the discussion are presented with respect to particular sampling locations there is no spatial information on the locations of these sampling locations (I have seen the coordinates in the tables, but this doesn't help/ would take a lot of time to locate the different stations on the map).

Reply 4: On the one hand, we focus on understanding of differences of groundwater residence times, recharge mechanisms and the interactions between groundwater and surface water in different districts with diverse hydrogeological settings rather than some locally particular locations based on the tracers' data to answer the research question in this study. On the other hand, the locations of samples taken at three farms of HH, QF and QS which were used for comparative analysis are presented schematically in Fig. 1.

Comment 5: Methods There is no clear methodology/procedure which explains how the results of the chemical analysis are treated. This also marks the big lack in this paper: There is no quantitative analysis of observed concentrations. For example, the presented nitrate and calcium concentrations are only "analyzed" with a rather surficial interpretation of "concentration groups" which does not fit at all (see comments on the result section). The method section would need a clear concept how the results of the chemical analysis were sorted, ranked, correlated, . . . and a hypothesis how this procedure will lead to the answers sought by this paper.

Reply 5: The answer to our problem in this study is more dependent on isotopic data. The hydrochemical data are combined with isotopic data to determine whether vertical recharge from surface water is occurring and to describe the water quality issues. Due to the lack of natural nitrate in most geologic formations, high nitrate concentrations generally indicate contamination by fertilizers from agricultural activities. The hydrochemical concentrations in different areas were used to determine where the interactions between groundwater and surface water and vertical recharge are occurring. Our chemical data may serve as baseline for future studies, especially on the water quality changes. See also "Reply 6" as follows.

Comment 6: While the results of the isotopic analysis show the overlay of surface and groundwaters the interpretation of the major ion concentrations does not fit at all: Figures 3 and 4 show similar concentrations for Districts II and III either for nitrate and for calcium and not as presented for District I and III. Consequently, the following interpretations should be reassessed.

Reply 6: Isotopic compositions of groundwater in the unconfined area (Districts I and III) are more enriched than that of groundwater in the confined area (District II) (Fig. 7). Most of the groundwater samples in the confined aquifers are plotted on the LMWL, while some groundwater in the unconfined aquifers are on the local evaporation line, indicating groundwater in the unconfined aquifers is more easily recharged by evaporated surface water with more enriched isotopic compositions. The groundwater samples at HH and QF farms (District II) (Fig. 8) are more depleted in heavy isotopes than the surface water, further indicating that lateral groundwater flow from mountainous area dominates the groundwater recharge, and that interactions with surface water barely occur. Groundwater at QS farm (District III) shows a wide range of tritium levels (<1.0-71.3TU), corresponding to the different sampling locations. Samples with high levels of tritium (6.5-71.3TU) are from shallow groundwater collected near the river, indicating strong links with river water. In District I, with strong links with surface water, groundwater nitrate concentrations can reach 458mg/L, while in the confined aquifer (District

II) with weak links with surface water, nitrate in most of the samples is less than 10mg/L (Fig. 3). The distribution of groundwater Ca2+ behaves similarly to groundwater nitrate (Fig. 4). The high Ca2+ values are derived from the soil layer with high Ca2+ contents by leakage of surface water in District I. Some samples with high NO3- and Ca2+ concentrations in the confined aquifer may indicate leakage from the shallow unconfined groundwater as groundwater table is continuously declining. The low NO3- concentrations in District III may be attributed to less fertilizers used in District III than in District I and "wetlands function" as discussed in section 5.3.

Comment 7: The Deuterium enrichment in the paddy field water samples is interpreted as condensation (Figure 6). This is wrong, condensation fractionates along the LMWL (saturated conditions). A possible reason might be methanogenesis which can cause heavy Deuterium-enrichment of soil water.

Reply 7: The water formed by condensation of evaporated moisture will also locate above the LMWL (Pang et al., Processes affecting isotopes in precipitation of an arid region, Tellus B, 2011, 63(3): 352-359.). We agree with you that methanogenesis may also be the reason which can cause heavy Deuterium-enrichment of soil water (Chidthaisong et al., 2002, Geochimica et Cosmochimica Acta, 66(6), 983-995.).

Comment 8: The discussion section misses for the largest parts the reflection of the actual literature with the results and the determination how the results presented within this study contribute to our understanding of the governing processes. For example, there is no explanation how the results of the groundwater age dating correspond to the various major ion concentrations (e.g. nitrate) for the different aquifer types and which recharge processes could cause observable chemical groundwater compositions.

Reply 8: The 2H and 18O isotopic compositions and 3H values in the unconfined area indicate that groundwater has strong interactions with surface water and modern recharge occurred, while groundwater in the confined area is recharged dominantly by lateral flow from the mountains around (discussed in Reply 6). So the groundwater age

in the confined area is generally older than that in the unconfined area. The vertical infiltration recharge from surface water with high nitrate from agricultural activities resulted in the generally high nitrate concentration in the unconfined area. However, it is not yet a serious problem in this area so should not be over emphasized, just to keep our argument well-ballanced.

―――――――――――――――――――――